# SPARSE WATERMARKING IN LLMS WITH ENHANCED TEXT QUALITY

## ABSTRACT

With the widespread adoption of Large Language Models (LLMs), concerns about potential misuse have emerged. To this end, watermarking has been adapted to LLM, enabling a simple and effective way to detect and monitor generated text. However, while the existing methods can differentiate between watermarked and unwatermarked text with high accuracy, they often face a trade-off between the quality of the generated text and the effectiveness of the watermarking process. In this work, we present a novel type of LLM watermark, *Sparse Watermark*, which aims to mitigate this trade-off by applying watermarks to a small subset of generated tokens distributed across the text. To demonstrate this type of watermark, we introduce **SpARK**, a **Sp**arse Waterm**ARK** method that achieves sparsity by anchoring watermarked tokens to words that have specific Part-of-Speech (POS) tags. Our experimental results demonstrate that the proposed watermarking scheme achieves high detectability while generating text that outperforms previous LLM watermarking methods in quality across various tasks.

## 1 INTRODUCTION

Recent advancements in Large Language Models (LLM) have shown exceptional performance in a multitude of tasks. From generating documents to answering questions on different topics, LLMs such as Meta's Llama (Touvron et al., 2023) and OpenAI's GPT (OpenAI, 2023) have become the foundation upon which many AI applications are built (Luo et al., 2023; Brohan et al., 2023; Luo et al., 2024; Huang et al., 2023). However, as these applications increase in their capabilities and accessibility, a growing risk of them being used for malicious purposes, such as generating fake news and being used for cheating assignments, becomes increasingly apparent.

With the ever-increasing problem of LLMs being misused, monitoring the generated text and its usage has become an increasingly crucial direction for research. One effective way for tracking the usage of generated text is by watermarking (Kirchenbauer et al., 2023; 2024; Zhao et al., 2024) - embedding imperceptible information into the generated text, thereby making it easier to detect and track for potential misuse. Recent studies have demonstrated the effectiveness and versatility of watermarks in embedding ownership information into generated text and distinguishing it from non-watermarked and human-written text (Krishna et al., 2023).

In addition to distinguishing between watermarked and non-watermarked texts, watermarking methods must also preserve the original text quality after embedding the secret information. However, prior works generally agree that there is a trade-off between the quality of the watermarked text and the strength of its watermark. For instance, Kirchenbauer et al. (2023) illustrates this trade-off by introducing a parameter that adjusts the extent to which their method affects the model's logits. By tuning this parameter, they demonstrate the balance between the quality of the generated text and the robustness of the watermark.

In this paper, we aim to circumvent the trade-off between watermark strength and text quality by proposing a watermarking method that augments only a portion of the generated text and checks for that portion of the text for watermark information. The main concept is illustrated in Figure 1. We show that by watermarking only a subset of the generated text, we can still maintain high detectability while minimizing the watermark's impact on the text quality. Our hypothesis is that while prior methods verify a watermark by checking every token within a text, the same effect can be achieved

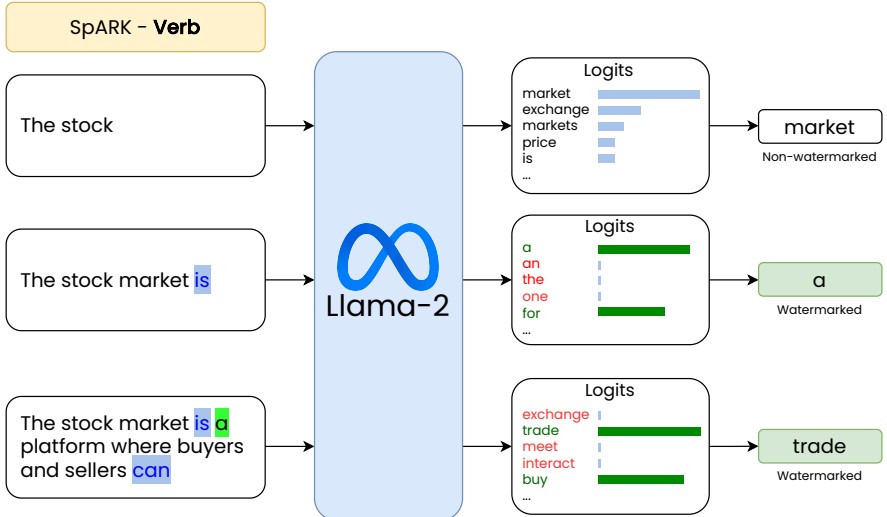

Figure 1: An overview of our proposed *SpARK*. For each generation step $t$, if the previous word belongs to the POS of interest (i.e., Verb), we divide the vocabulary into *Green/Red* list and restrict sampling from the *Green* list. Otherwise, we generate the next token with the original probability.

by checking only a specific portion if the locations of the watermarked elements are known. This helps preserve the quality of the generated text by keeping a large portion of the original generated text while still successfully embedding the secret information. Our contributions can be summarized as follows:

- We introduce *Sparse Watermark*, a novel category of watermarking methods for LLMs that are designed to preserve both text quality and detectability by selectively watermarking and verifying only a subset of the generated text.

- We propose *SpARK*, a method of watermarking using Part-of-Speech (POS) tags, embedding and detecting watermarks based on the POS tags of words within the generated text.

- Through extensive experiments on *SpARK*, we demonstrate that *Sparse Watermark* effectively maintains high text quality generated by LLMs and watermark detectability, outperforming several previous methods across various generation tasks.

## 2 RELATED WORKS

**AI-generated text detection.** The methods for monitoring the usage of AI-generated text can be generally classified into two main categories: AI text detection and watermarking. Of these, watermarking has proven to be more reliable and effective for distinguishing between generated and human-written text, as well as between watermarked and unwatermarked generated text (Krishna et al., 2023). In addition, as companies and research communities strive to close the gap between LLM-generated and human-written texts, relying solely on AI text detection of the original text will become increasingly challenging. The main objective of LLM watermarking is to inject secret information imperceptible to humans into the generated text, which can later be verified by using watermark detection mechanisms (Kirchenbauer et al., 2024; Zhao et al., 2024; Kirchenbauer et al., 2024; Liu et al., 2024a; Gu et al., 2024).

**Text watermarking for LLM.** One common approach of text watermarking for LLMs focused on distorting the next token probability distribution of the language model. This is achieved by randomly dividing the vocabulary into two disjoint sets named *Green* list and *Red* List, and then promoting the generation of only tokens in the *Green* list with a bias parameter $\delta$ (Kirchenbauer et al., 2023). During detection, a detector with the secret key could recover the watermarked distribution and use a statistical test to verify the presence of the watermark.

Recent works have attempted to improve LLM watermarking from the perspective of advancing robustness and security. For instance, Zhao et al. (2024); Kirchenbauer et al. (2024) explored various schemes to enhance the robustness of watermarks. Zhao et al. (2024) illustrated that leveraging a fixed *Green* list enabled watermarking to be resilient against various types of attacks. Kirchenbauer et al. (2024) explored several hashing schemes for improved robustness. Training-based watermarks are also designed, where one study improved the robustness of the watermark using the semantics of previously generated tokens (Liu et al., 2024b). Liu et al. (2024a) proposed to train two neural networks for text generation and watermark detection to create an unforgeable watermark. Lee et al. (2023) introduced entropy thresholding for code generation as watermarking low-entropy tokens could compromise the correctness of the generated sequences.

**Effects of watermark on text quality.** However, while these recent works have considerably enhanced the robustness, detectability, and unforgeability of LLM watermarking, it is generally agreed that there is a trade-off between the quality of the watermarked text and the strength of its watermark. The distribution shift introduced in Kirchenbauer et al. (2023) enhances the detectability of the watermark, but it simultaneously allows less likely tokens to be generated, thus affecting the intrinsic quality of the generated text. Recently, Tu et al. (2023) introduced a benchmark method of several LLM watermarking algorithms and verified this deterioration of text quality. To minimize the impact on the generation quality, Christ et al. (2023); Kuditipudi et al. (2023) proposed to embed the watermark during the token sampling process, thus inducing zero distortion to the probability distribution of the LLM. However, in practice, the sampling-based schemes struggled to produce a detectable watermark for low-temperature settings (Piet et al., 2023). Huo et al. (2024) introduced a multi-objective optimization method to dynamically generate bias parameters and *Green* list ratio to achieve both detectability and semantic coherence. In contrast, our approach, *SpARK*, emphasizes preserving the strength and semantic integrity of generated text by leveraging the innate structure of natural language, eliminating the need for training.

## 3 PROPOSED METHOD

### 3.1 NOTATIONS AND PRELIMINARIES

We first introduce the notations used in this paper. Let $\mathcal{M}$ be an autoregressive language model that takes a tokenized prompt $\mathbf{x}_{\text{prompt}} = \{x_{-N}, ..., x_{-2}, x_{-1}\}$ and output a sequence of tokens that simulate natural responses. At generation step $t$, the input for the language model $\mathcal{M}$ is combined sequences of tokens $\mathbf{x}_{\text{prompt}}$ and the tokens $\mathbf{x} = \{x_0, ..., x_{t-1}\}$ previously generated by $\mathcal{M}$ in the previous steps. The language model $\mathcal{M}$ then takes the input and outputs a probability distribution of the next token over the vocabulary $\mathcal{V}$ of the language model: $P_{\mathcal{M}}(x_{-N}, ..., x_{t-1}) = (P_{\mathcal{M}}(v|x_{-N}, ..., x_{t-1})|v \in \mathcal{V})$.

According to Kirchenbauer et al. (2024), watermark algorithms are defined using four parameters. The hash function $\mathcal{H}$ generates a pseudo-random $hash$ using the context of the generated text with context width $h$, the fraction of green list token $\gamma$, and the magnitude of the logit bias $\delta$. After the watermarked text is generated, one can use the same parameters to calculate and retrieve a set of green tokens $s$ in the generated text. We then use this set to calculate the statistical significance of $|s|$ number of green tokens that appeared in the generated text with token length $T$. We can use a one-proportion z-test assuming the null hypothesis $\mathcal{H}_0$ which states: "The text sequence is generated without a watermark". The z-score is then calculated as

$$z = \frac{|s| - \gamma T}{\gamma \sqrt{(1 - \gamma)T}}. \tag{1}$$

If a text sequence's $z$-score surpasses a defined threshold, we can confidently determine that the text has been watermarked.

### 3.2 THREAT MODELS

In this paper, we consider the same threat model as in prior works (Kirchenbauer et al., 2023; Zhao et al., 2024; Liu et al., 2024a). The goal is to embed a watermark for LLM so that users can later verify if certain texts are generated by the LLM. We assume that the adversary is aware of the presence of watermarks and attempts to evade the watermark detection when using the LLM. The adversary

---

**Algorithm 1** Text Generation with SpARK

---

1: **procedure** GENERATETEXT($\mathbf{x}_{\text{prompt}}$)
2:     **for** $t = 0, 1, \ldots$ **do**
3:         $P_{\mathcal{M}}(t) \leftarrow \mathcal{M}(x_{-N}, \ldots, x_{t-1})$
4:         $hash \leftarrow \mathcal{H}(x_{-N}, \ldots, x_{t-1})$
5:         $P_{\mathcal{M}}(t) \leftarrow \text{POSWatermark}(\mathbf{x}_{\text{prompt}}, P_{\mathcal{M}}(t), hash)$
6:         $\mathbf{x}_{\text{prompt}}(t) \leftarrow \text{Sample}(G)$
7:     **end for**
8:     **return** $\mathbf{x}_{\text{prompt}}[0 : t]$
9: **end procedure**

---

**Algorithm 2** SpARK Encoding

---

1: **procedure** POSWATERMARK($\mathbf{x}$, $P_{\mathcal{M}}$, $hash$)
2:     $T \leftarrow$ Convert tokens $\mathbf{x}$ to normal text
3:     $W \leftarrow$ Last word of $T$
4:     $P_{\text{tag}} \leftarrow \text{POS}(W, T)$
5:     **if** $P_{\text{tag}} \in I$ **then**
6:         $G \leftarrow \text{GenerateGreenList}(T, hash)$
7:         $P_{\mathcal{M}} \leftarrow \text{ApplyGreenList}(P_{\mathcal{M}}, G)$
8:     **end if**
9:     **return** $P_{\mathcal{M}}$
10: **end procedure**

---

could have access to both open-source and private (non-watermarked) language models to produce an alternate text. Consistent with prior works, we only consider attacks such that the modifications are able to erase the watermark without significantly deviating from the original semantics of the texts.

### 3.3 SPARSE WATERMARKING USING POS TAGS

In previous works, most watermarking techniques attempt to encode watermark information into each token in the generated text. As the strength of the watermarking method increases, more tokens are adversely affected, which decreases the quality of the generated text (Kirchenbauer et al., 2023). We aim to improve the text quality by watermarking the generated text sparsely, which however is non-trivial. Attempting to watermark sparsely without knowing the location of the watermarked elements would be akin to using the previous watermark methods with low strength. This is due to the statistical test also including the non-watermarked portions of the generated text. To this end, by isolating and conducting the statistical test specifically on the watermarked portions of the generated text, we can significantly enhance detectability while maintaining higher text quality compared to using previous methods with stronger watermarking.

We utilize the Universal Part-of-Speech (POS) tags (Petrov et al., 2012) that exist in the generated text to mark the positions of the watermarked tokens in the text sequence. Specifically, during the generation process, we select the positions to watermark based on the POS of tokens that have been generated, allowing the watermark positions to be tied to the sentence structure. This makes the watermark more resilient to insertions/deletions of tokens in the generated text and also makes it easier to extract the watermarked portion of the text using the POS tags.

Before using SpARK, we first select a list of POS tags $I$ to be used for watermarking. When the text generation process starts, as described in Algorithm 1, we verify when the model has generated a full word by determining if the next token with the highest probability is the start of a new word. While LLMs sample the next token differently with different sampling schemes, using this strategy could consistently inform us when a full word has been generated, without the need to backtrack during the generation process. Once a full word is produced by the language model, we obtain its POS tag $P_{tag}$, and watermark the next token only if $P_{tag} \in I$. We choose to watermark the token next to the word with chosen POS tags, as watermarking those words directly would not guarantee it to have the same POS tag after being watermarked, leading to inconsistencies. By using words that have a selected POS as an anchor, we can limit the number of watermarked tokens in the generated text and position

---

**Algorithm 3** SpARK Watermark Detection

---

**procedure** DETECTWATERMARK($\mathbf{y}, I, hash$)
    $s = 0$
    $T = 0$
    **for** $i = 1, 2, ... |\mathbf{y}|$ **do**
        $P_{tag} \leftarrow \text{POS}(\mathbf{y}[i], \mathbf{y}[: i])$
        **if** $P_{tag} \in I$ **then**
            $T = T + 1$
            $next\_token \leftarrow \text{NextToken}(\mathbf{y}[: i], i)$
            $G \leftarrow \text{GenerateGreenList}(\mathbf{y}[: i], \text{hash})$
            **if** $next\_token \in G$ **then**
                $s = s + 1$
            **end if**
        **end if**
    **end for**
    $z \leftarrow \frac{s - \gamma T}{\gamma \sqrt{(1 - \gamma) T}}$
    **if** $z > $ threshold **then**
        **return** True
    **else**
        **return** False
    **end if**
**end procedure**

---

them to be easily relocated when decoding. We outline the process of watermarking using POS tags in Algorithm 2.

To watermark the next token, we used a similar process and hashing scheme as described in Kirchenbauer et al. (2023), partitioning the vocabulary using $\gamma$ and limiting the generations of new tokens to a subset of the vocabulary, the *Green* list $G$. While dividing the vocabulary, we only select tokens that start a new word, as it would not affect the previous words and their POS tags, making the decoding process more consistent. Additionally, our method does not use $\delta$ to increase the bias for generating green list tokens, but instead, we restrict the model to only select from the *Green* list. This helps the encoding process to utilize all of the tokens it has access to, as it can only watermark a small portion of the generated text.

### 3.4 SpARK WATERMARK DETECTION

Since SpARK watermark is sparse, we identify the specific positions we have selected for the watermark to ensure that the unwatermarked portions of the text are not considered in the z-score calculation. This process would preserve the strength of the sparse watermark. We first identify the words whose POS tags are in the list $I$ and select the next token. These selected tokens are the ones we would watermark during the encoding process and thus would be in the *Green* list $G$. At each of the selected token positions, we recover the $G$ using the hashing scheme mentioned in the encoding process and check if the token in that position is in $G$. We then calculate the statistical significance of the number of green tokens that appeared in the generated text, as shown in Equation 1. However, as we only apply the watermark to tokens after the words with a specific POS, $T$ (the total number of tokens) would be replaced by the number of tokens in the watermarked positions. The watermark detection step is presented in Algorithm 3.

## 4 EXPERIMENTS

### 4.1 EXPERIMENTAL SETUP

We choose Llama2-7b, a popular open-sourced LLM that has been instruction-tuned to align with human preference, as our baseline model for testing the watermarking methods. In addition, we also conduct the experiments on Phi-3, a 3.8 billion language model that has been shown to outperform

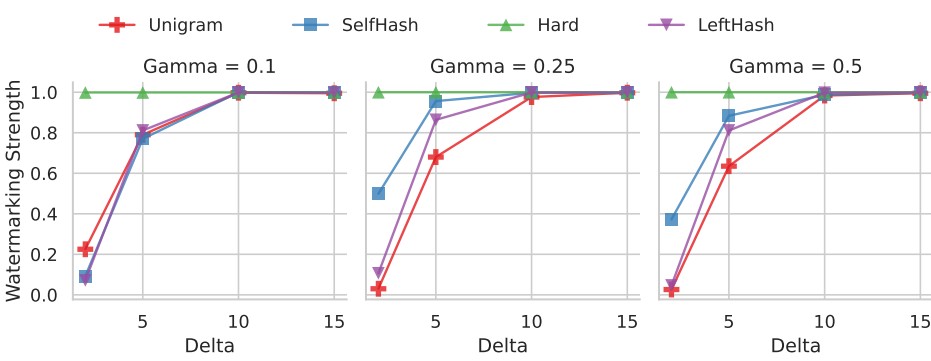

Figure 2: True Positive Rate (TPR) for each method on the selected dataset, generated using Llama2-7b with different hyper-parameters for $\gamma$ and $\delta$.

bigger LLMs on several benchmarks. We compare the performance of our proposed method against four LLM watermarking techniques:

- **Hard watermark:** The initial watermark method proposed by Kirchenbauer et al. (2023). This method restricts the model to only generating a portion of the vocabulary, referred to as the *Green* list, and uses the statistical test to detect the watermark.

- **LeftHash watermark (Soft watermark):** A watermark method proposed by Kirchenbauer et al. (2023). The method is similar to Hard watermark, but this watermark encourages the model to generate tokens from the *Green* list by adding a constant $\gamma$ to the output logit, instead of restricting the model. We refer to this method as LeftHash to differentiate this method from another method proposed in Kirchenbauer et al. (2024).

- **SelfHash watermark:** A watermark method proposed by Kirchenbauer et al. (2024). This watermark method is similar to LeftHash watermark, as it encourages the model to generate from the *Green* list as well. The main difference is that this watermarking method chooses tokens that contain themselves in the *Green* list during hashing, increasing robustness.

- **Unigram watermark:** A watermark method that simplifies the watermark process by utilizing a fixed *Green* list used to watermark text (Zhao et al., 2024). Their work shows that this restriction increased the robustness of the watermark.

To validate the detectability and the quality of the text generated by the watermark methods, we used an experiment setting similar to WaterBench (Tu et al., 2023). This benchmark procedure aims to measure both the quality of the generated text and its detectability. We focused the experiment only on the long-answer datasets, following the same setting as in prior works of LLM watermarking (Kirchenbauer et al., 2024; Gu et al., 2024). We conduct the same hyper-parameter search experiments on long-answer datasets to find parameters that are more suitable to watermark these long text answers. The watermarking strength results are shown in Figure 2. During the hyper-parameter search, we select the parameters that are close to the original parameters of each method and have a True Positive Rate of greater than 0.99.

To summarize, we select the ELI5 (Explained Like I'm 5) dataset (Fan et al., 2019) and the FinanceQA dataset (Maia et al., 2018), both of which focus on short questions with long answers, along with MultiNews (Fabbri et al., 2019) and QMSum (Zhong et al., 2021), which focuses on text summarization. These four datasets are grouped into two tasks, Long-form QA and Summarization. We then conduct a hyper-parameter search by evaluating the TPR of each method using different hyper-parameters. As shown in Figure 2, the strength of the watermark increases as $\gamma$ decreases and $\delta$ increases, The figure also shows that most watermark methods achieved over 0.99 of TPR if $\delta$ is high enough, which helps us choose a $\gamma$ that is close to the original parameters of each method. We then select hyper-parameters closest to the original paper, while having a TPR of over 0.99.

For SpARK, we selected three POS tags for the main experiment: Verb, Noun, and Determiner. This is because, based on Table 7, these three tags have 100% of document frequency. We selected $\gamma = 0.05$ for the SpARK, to increase the strength of each watermark toke. By choosing a small $\gamma$,

Table 1: Comparison of True Positive Rate (TPR), True Negative Rate (TNR), ROUGE-L score (R-L), decrease in percentage point of ROUGE-L score ($\Delta$) and the semantic similarity of watermarked and non-watermarked text (Sem.) of different watermarking algorithms, evaluated on Llama-2 model. The best and second-best performances are in **bold** and underline, respectively.

| | Long-form QA | | | | Summarization | | | | Sem. |
|---|---|---|---|---|---|---|---|---|---|
| | TPR | TNR | R-L | $\Delta$ | TPR | TNR | R-L | $\Delta$ | |
| No Watermark | – | – | 21.59 | – | – | – | 23.47 | – | – |
| Hard | 100.0 | 100.0 | 16.76 | ↓ 22.37% | 100.0 | 100.0 | 16.63 | ↓ 29.14% | 0.765 |
| LeftHash | 100.0 | 100.0 | 14.55 | ↓ 32.61% | 99.5 | 99.5 | 13.33 | ↓ 43.20% | 0.693 |
| SelfHash | 99.5 | 93.5 | 12.75 | ↓ 40.94% | 100.0 | 96.0 | 12.54 | ↓ 46.57% | 0.663 |
| Unigram | 99.8 | 100.0 | 11.43 | ↓ 47.06% | 99.3 | 100.0 | 11.53 | ↓ 50.87% | 0.652 |
| SpARK - Verb | 100.0 | 99.0 | 18.87 | ↓ 12.60% | 100.0 | 99.5 | **20.95** | ↓ 10.74% | **0.836** |
| SpARK - Noun | 100.0 | 99.5 | 18.48 | ↓ 14.40% | 100.0 | 100.0 | 18.39 | ↓ 21.64% | 0.794 |
| SpARK - Determiner | 100.0 | 98.8 | **19.20** | ↓ 11.07% | 100.0 | 98.0 | 20.89 | ↓ 10.99% | 0.814 |

Table 2: Comparison of True Positive Rate (TPR), True Negative Rate (TNR), ROUGE-L score (R-L), decrease in percentage point of ROUGE-L score ($\Delta$) and the semantic similarity of watermarked and non-watermarked text (Sem.) of different watermarking algorithms evaluated on Phi-3 model. The best and second-best performances are in **bold** and underline, respectively.

| | Long-form QA | | | | Summarization | | | | Sem. |
|---|---|---|---|---|---|---|---|---|---|
| | TPR | TNR | R-L | $\Delta$ | TPR | TNR | R-L | $\Delta$ | |
| No Watermark | – | – | 22.62 | – | – | – | 23.37 | – | – |
| Hard | 100.0 | 100.0 | 15.19 | ↓ 32.83% | 100.0 | 100.0 | 11.22 | ↓ 52.01% | 0.567 |
| LeftHash | 100.0 | 100.0 | 19.55 | ↓ 15.34% | 99.3 | 99.5 | 15.71 | ↓ 32.78% | 0.779 |
| SelfHash | 100.0 | 97.0 | 19.51 | ↓ 13.75% | 99.8 | 99.5 | 16.85 | ↓ 27.90% | 0.806 |
| Unigram | 100.0 | 100.0 | 7.74 | ↓ 65.77% | 99.8 | 100.0 | 7.04 | ↓ 69.88% | 0.425 |
| SpARK- Verb | 100.0 | 99.0 | **21.45** | ↓ 5.17% | 100.0 | 99.5 | **20.87** | ↓ 10.72% | **0.850** |
| SpARK- Noun | 99.5 | 99.5 | 19.46 | ↓ 13.95% | 100.0 | 100.0 | 18.27 | ↓ 21.84% | 0.787 |
| SpARK- Determiner | 99.5 | 98.8 | 21.18 | ↓ 6.37% | 100.0 | 99.0 | 20.86 | ↓ 10.74% | 0.829 |

we demonstrate that SpARK,and Sparse Watermark in general, can match other baseline methods in detectability, while also maintaining higher generation performance. We also provide the results containing the TPR of SpARK under different POS tags and $\gamma$.

## 4.2 RESULTS OF DETECTABILITY AND TEXT QUALITY

As mentioned in Section 4.1, we conduct evaluations with parameters that achieved greater than 0.99 TPR and close to the original parameters of each method. We report the results of the watermarks' performance in Table 1 and Table 2.

Overall, the detection performance of the baseline watermark method is high, having above 99% True Positive Rate and True Negative Rate in both tasks. In addition, tuning the parameter for long-answer text increased the generation performance of all watermark methods without degrading their detectability. Compared to the baseline watermark methods, our SpARK achieved similar detection performance, while consistently achieving the highest generation performance in both tasks. On Llama2-7b, all three of the SpARK variants using different POS tags reached the top 3 spots in terms of generation performance. When using any of SpARK variants, the ROUGE-L score of the original model would only be reduced by at most $21.64\%$. In contrast, the performance of other watermarks would decrease that down by at least $22\%$ and at most more than $50\%$. SpARK also has the highest semantic similarity between the non-watermarked text and watermarked text, with the highest being $0.836$ and the lowest being $0.794$.

The same phenomenon can be seen on Phi-3, as SpARK maintains the generated text quality while having a high TPR compared to other methods. For long-form QA, the Verb and Determiner variants of SpARK only reduce the quality by roughly $5\%$ and $6\%$, respectively, while other baseline

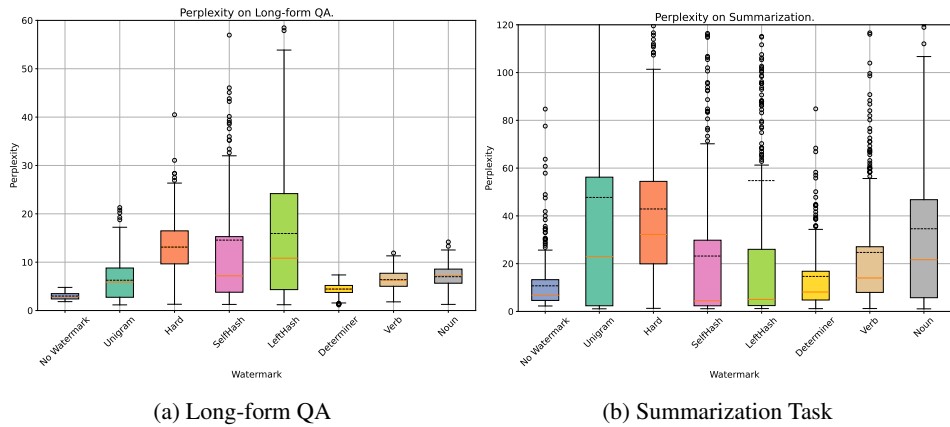

(a) Long-form QA  (b) Summarization Task

Figure 3: Perplexity of the generated text for each watermark method in two main tasks.

watermark decreases the quality of the text by at least $13.75\%$ (SelfHash) and at most $65.77\%$ (Unigram). For summarization, SpARK maintains the best generation text with all three variants, degrading the quality by at most $21.8\%$, where other methods admit decreases of at least $27\%$. In terms of semantic similarity between watermarked and non-watermarked text, SpARK maintains the highest positions, with the Verb variant achieving $0.850$ and Determiner $0.829$.

We also plot the perplexity of each watermark method's responses to measure the generation quality. We used Llama2-13b as the oracle model to measure perplexity, as it is a more powerful language model that is publicly available, similar to the methodology in Jovanović et al. (2024). As shown in Figure 3, SpARK consistently achieves the lowest perplexity across both tasks when using the Determiner POS tag. Notably, our method also induces lower variance in perplexity, indicating that it not only maintains low perplexity but does so with greater consistency, emphasizing the stability of our approach compared to existing watermarking techniques.

In summary, SpARK's results show that sparse watermark can produce better-generated texts, both in terms of semantic similarity and task performance, while having the same detectability. By watermarking a small portion of text sparsely and anchoring each watermarked token with a POS tag, SpARK can preserve the performance and similarity of the generated text, while maintaining detectability by focusing the detection on smaller sets of tokens.

### 4.3 RESULTS OF ROBUSTNESS AGAINST ATTACKS

As malicious players have the capability of modifying a sequence of watermarked text to evade the detector, watermarking methods need to ensure that the watermark is resilient against changes to the text. In order to illustrate the robustness of our proposed approach, we consider two realistic types of attack: substitution attack and paraphrasing attack.

**Substitution Attack.** For the substitution attack, a specified proportion of the text (equal to some $r$ tokens) is replaced with its corresponding synonyms. However, it is worth noting that a simplistic replacement can compromise the semantic coherence of the sentence. Following the settings described in (Wang et al., 2024), we iteratively masked a random token that has yet to be modified and then utilized RoBERTa-Large to generate candidates for replacement. To ensure the semantic integrity of the perturbed text, we only select to substitute a new token if the difference in logits of the new token and the original is higher than our pre-defined threshold, which we set to be -1. If there is no token that satisfies the preceding requirement, we proceed to mask a different token. The process is terminated when we have replaced $r$ tokens or we have attempted to replace $3r$ tokens.

Table 3 demonstrates the resilience of our method against substitution attack, with SpARK achieving good performance for the $10\%$ scenario. For higher rates such as $30\%$, the robustness of our proposed method lessens, but they remain competitive with other watermarking algorithms. Detailed results of each dataset for the substitution attack can be found in Table 10 and Table 11 of the Appendix.

**Paraphrasing Attack.** In addition to the substitution attack, we also evaluate the robustness of our proposed method against paraphrasing attack using DIPPER (Krishna et al., 2023). DIPPER is an

Table 3: Average True Positive Rate under two settings of attacks: synonym substitution and paraphrasing (DIPPER), evaluated on Llama2-7b and Phi-3. The best and second-best performances are in **bold** and underline, respectively.

| Language Model | Method | Substitution Attack | | | DIPPER | |
|---|---|---|---|---|---|---|
| | | 10% | 30% | 50% | 40L | 40L-40O |
| Llama2-7b | Hard | 99.6 | 90.6 | 51.1 | 53.0 | 41.0 |
| | LeftHash | 99.8 | **99.0** | 83.9 | 71.4 | 64.1 |
| | SelfHash | 99.8 | 98.1 | **92.3** | **75.0** | **69.5** |
| | Unigram | 99.5 | 96.9 | 91.4 | 59.8 | 50.9 |
| | SpARK- Verb | 99.8 | 96.3 | 72.4 | 54.3 | 43.5 |
| | SpARK- Noun | **100.0** | 97.8 | 78.3 | 53.9 | 41.9 |
| | SpARK- Determiner | 99.8 | 96.5 | 67.6 | 74.3 | 66.9 |
| Phi-3 | Hard | **100.0** | **100.0** | 98.6 | **89.3** | 88.3 |
| | LeftHash | 99.3 | 98.1 | 83.6 | 79.8 | 66.1 |
| | SelfHash | 99.3 | 96.1 | 62.8 | 79.5 | 66.9 |
| | Unigram | 99.9 | 99.6 | **99.3** | 89.1 | **88.6** |
| | SpARK- Verb | 99.6 | 96.3 | 72.5 | 64.0 | 54.1 |
| | SpARK- Noun | 99.4 | 97.5 | 80.6 | 71.5 | 59.1 |
| | SpARK- Determiner | 99.6 | 96.8 | 76.6 | 87.1 | 82.4 |

11B parameter model that has been specially fine-tuned from T5-XXL (Raffel et al., 2020) for the task of paraphrasing. It has been demonstrated to successfully evade multiple AI-generated text detectors while also preserving the general semantics of the sentence. We assess the performance of our watermarking schemes for two attack settings: 40L, where the lexical diversity is set to 40, and 40L-40O, where the lexical and order diversity are 40. With these configurations, DIPPER is able to produce a strong paraphrasing attack and maintain a high degree of semantic similarity with the original sentence.

The results of the paraphrasing attacks are summarized in Table 3. When applied to Llama2-7b, the performance of SpARK with determiner is demonstrated to be near state-of-the-art in terms of robustness, achieving $74.3\%$ and $66.9\%$ in true positive rate, only $0.7$ and $2.6$ percentage points behind SelfHash, under 40L and 40L-40O paraphrasing, respectively. For Phi-3, SpARK can still achieve $87.1\%$ and $82.4\%$ for DIPPER, higher than both LeftHash and SelfHash. While Hard watermark and Unigram achieved higher robustness on Phi-3, their generated texts have the lowest scores compared to other methods, as shown in Table 2. In contrast, SpARK was able to achieve relatively high robustness against attacks while having the best results in terms of generated text on both Llama2-7b and Phi-3. The performance of each method for all datasets can be found in Table 12 and Table 13 of the Appendix.

## 4.4 EMPIRICAL EFFECTS ON Z-SCORE AND TEXT QUALITY

To demonstrate SpARK's ability to maintain both high detectability and preserve the semantic meaning of the non-watermarked generation, we provide an example of watermarking applied to an answer in QMSum using SelfHash and SpARK - Determiner. This table visually demonstrates the watermarked tokens and their corresponding list, with tokens found in the *Red* list represented in red, and tokens found in the *Green* list represented in green. As we can observe in Table 4, techniques like SelfHash aim to watermark every token when generating, while SpARK only focuses on watermarking only a fraction of the generated tokens. While SelfHash has a large *Green* list with $\gamma = 0.25$, the quality of the text being generated by SelfHash has a lower similarity, only $(0.298)$ due to the number of tokens it encodes. SpARK, on the other hand, even when having a smaller green list ($\gamma = 0.05$), the generated text has a higher semantic similarity than SelfHash $(0.726)$, thanks to encoding fewer tokens. While SelfHash's generated text does have a higher $z$-score $(16.99)$ compared to SpARK's $11.53$, it is worth emphasizing that the number of tokens used in SpARK is a lot smaller. SpARK is able to maintain a similar level of detectability to SelfHash as seen in Section 4.2. Additional watermarked text examples from different watermark methods in different datasets can be found in Section K of the Appendix.

Table 4: Examples from the QMSum dataset generated by Llama2-7b with no watermark (NW), with SelfHash, and with SpARK - Determiner (SPK-Det), respectively. Results of z-score and semantic similarity (Sem.) of each watermarking method are also reported. The hyper-parameter $(\gamma,\delta)$ for SelfHash is $(0.25, 10)$, and the hyper-parameter $\gamma$ for SpARK is $0.05$.

| Prompt | [INST]You are given a meeting transcript and a query containing a question or instruction. Answer the query in one or more sentences.\n\nTranscript:\nUser Interface: ...\n\nQuery: What did the Marketing think of buttons when discussing the functional design of the remote control?\nAnswer:[/INST] | $z$-score | Sem. |
|---|---|---|---|
| NW | Based on the meeting transcript, the Marketing person suggested that there should be no buttons on the remote control, as most people only use a small number of buttons on their existing remote controls, and having too many buttons can be confusing. Instead, they proposed using a menu-based system or a single button for a menu. | - | - |
| SelfHash | At the meeting, the marketing person suggested eliminating unnecessary buttons on the remote control aimulating a more user-frienf design. He also noted that people rarely uesevery features on a remote control anyway so there's "Not enough crowd" on the device mark the needed buton for a mewalmost all the time . Therefore , the Markettion believed the onerouse was to conisder a more minimalinconvenie _ design f the product Marktng's view was to focus the user expdria and keep the device slew a d simple... ... | 16.99 | 0.298 |
| SPK-Det | Based on the extract, the design manager expressed that the design project team should aim to create a remote control with fewer buttons than usual, as most people do not use their TV remote controls' full capacity. The Designer also suggested that a minimalist approach could be beneficial, with only one button for a shortcut menu | 11.53 | 0.726 |

## 5 CONCLUSION

In this work, we propose *SpARK*, a novel watermark method for LLM, that encodes watermark information into the generated text, without degrading its quality. Different from other methods, this approach focuses on encoding a subset of tokens distributed sparsely throughout the generated text. By encoding a small subset of tokens in the generated text and focusing on those subsets for watermark detection, SpARK can minimize the impact of the watermark on the text quality while maintaining high detectability. Experimental results demonstrate the effectiveness of our *SpARK* in preserving the text quality, as evidenced by the ROUGE-L score and semantic similarity for four datasets compared to other methods. Despite watermarking significantly fewer tokens, our approach maintains competitive robustness against both substitution and paraphrasing attacks.

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

## A Theoretical support for methods

### A.1 Notations

- $\mathbf{x}_{prompt} = \{x_{-N}, \ldots, x_{-1}\}$: Tokenized prompt.
- $\mathbf{x} = \{x_0, \ldots, x_{t-1}\}$: Sequence of previously generated tokens.
- $\mathbf{y}$: Generated Text / Normal text.
- $P_{\mathcal{M}}(x_{-N}, \ldots, x_{t-1})$: Probability distribution of the next token generated by the language model $M$.
- $T$: Normal text converted from tokens $\mathbf{x}_{prompt}$ and $\mathbf{x}$.
- $W$: Last word in $T$.
- $P_{tag}(x)$: POS tag of word $x$.
- $I$: Set of selected POS tags for watermarking.
- $G$: Green List.

### A.2 Mathematical formulation

**POS Tag Selection:** Define a set of POS tags $I$ chosen for watermarking based on their frequency and relevance in the text. For example:

$$I = \{\text{DET}\}$$

In this work, we only select a single POS tag to watermark. This is because the tags we used, Universal POS tags, are composed of smaller POS tags defined by Penn Treebank. For example, the Universal tag DET when converted to Penn Treebank POS tags would look like so:

$$I = \{\text{DT}, \text{EX}, \text{PDT}, \text{WDT}\}$$

**Token Generation Process:** For each generation step $t$, the language model $\mathcal{M}$ generates a probability distribution over the vocabulary $\mathcal{V}$:

$$P_{\mathcal{M}}(x_{-N}, \ldots, x_{t-1}) = (P_{\mathcal{M}}(v \mid x_{-N}, \ldots, x_{t-1}) \mid v \in \mathcal{V})$$

**POS Tag Identification:** Determine the POS tag $P_{tag}(W)$ of the last word $W$ by using a POS parser:

$$P_{tag}(W) = \text{POS}(W, T)$$

Here, $\text{POS}(W, T)$ represents the function that returns the POS tag of the word $W$ based on its position in text $T$.

**Watermark Application:** If the POS tag $P_{tag}(W) \in I$, apply the Green List $G$ to the probability distribution, modifying it to embed the watermark:

$$\tilde{P}_{\mathcal{M}} = \text{ApplyGreenList}(P_{\mathcal{M}}, G)$$

The modified probability distribution $\tilde{P}_{\mathcal{M}}$ will be used to generate the next token.

**Green List Generation:** The function $\text{GenerateGreenList}(T, hash)$ generates the list of green tokens $G$ based on the current context $T$. This list is used to modify the probability distribution of the next token:

$$G = \text{ApplyGreenList}(T, hash)$$

## B Additional implementation details

### B.1 Hyper-parameters of baseline watermark methods

To obtain the hyper-parameters for each baseline watermark method to be used in the main experiment, we conduct the same hyper-parameter search explained in the original paper by Tu et al. (2023). We select the "Watermark Strength", which is the True Positive Rate to be 0.99 or above, as all baseline watermarks can achieve high watermarking strength while only adjusting $\delta$. This is because we tuned the hyper-parameters on only the long-answer dataset. All the hyper-parameters selected for the main experiments can be found in Table 5.

Table 5: Hyper-parameters for each baseline watermark method used in the main experiment.

| Llama2-7b | | | | |
|---|---|---|---|---|
| Parameters | Hard | LeftHash | SelfHash | Unigram |
| $\gamma$ | 0.5 | 0.25 | 0.25 | 0.5 |
| $\delta$ | - | 10 | 10 | 15 |
| $z$-threshold | 4.0 | 4.0 | 4.0 | 4.0 |
| Phi-3 | | | | |
| Parameters | Hard | LeftHash | SelfHash | Unigram |
| $\gamma$ | 0.25 | 0.25 | 0.5 | 0.25 |
| $\delta$ | - | 5 | 5 | 10 |
| $z$-threshold | 4.0 | 4.0 | 4.0 | 4.0 |

### B.2 HYPER-PARAMETER TUNING FOR SPARK

Table 6: True Positive Rate (TPR) of SpARK on different POS tags.

| POS Tags | Gamma ($\gamma$) | | | |
|---|---|---|---|---|
| | 0.05 | 0.1 | 0.25 | 0.5 |
| Verb | 100.00% | 100.00% | 97.63% | 77.38% |
| Noun | 100.00% | 99.88% | 98.50% | 85.75% |
| Determiner | 100.00% | 99.88% | 96.13% | 55.50% |
| Preposition and Postposition | 100.00% | 99.63% | 95.25% | 63.00% |
| Punctuations | 99.63% | 99.63% | 90.88% | 62.63% |
| Adjective | 97.75% | 94.25% | 77.38% | 22.38% |

To find the hyper-parameters for SpARK, we conduct the same hyper-parameter search process used for the baseline watermark. We decided only to conduct the search on POS tags that have a document frequency of 99% or over, as we want to use the POS tags that have a near guarantee of occurring in a document. From Table 6, we can see that Punctuations and Adjectives cannot give a 100% True Positive Rate, even with a gamma of 0.05. This is because the chance of adjectives appearing in a document is 99%, which can cause some generated samples to be unwatermarked. While punctuations have a high document frequency, a lot of answers only use punctuations at the end of the sentence, which causes the generated text to be unwatermarked as well.

From the True Positive Rate shown in Table 6, we selected $\gamma = 0.05$ for all of the SpARK variants in every experiment. By choosing this hyper-parameter, we show that SpARK can achieve high detection performance while having generation performance higher than all of the baseline methods, even while the True Positive Rate is at 100%.

### B.3 IMPLEMENTATION OF SEMANTIC SIMILARITY

In this paper, we measure the closeness in semantics of watermarked and non-watermarked texts to understand the semantic distortion of applying a watermark to an LLM. Results of semantic similarity in Table 1 are calculated by computing the cosine similarity between the embeddings produced by SimCSE (Gao et al., 2021) of texts generated with and without watermark. SimCSE leverages contrastive learning to train BERT (Devlin et al., 2019) and RoBERTa (Liu et al., 2019) models for generating sentence embeddings.

## C ANALYSIS ON THE POS TAGS

### C.1 DOCUMENT FREQUENCY OF THE POS TAGS

As our SpARK uses a POS tag to mark the positions of the watermarked text, not all POS tags would occur during the generating process. Because of this, we calculated the document frequency of each POS tag that appeared in the provided answers from the datasets. This is to show which POS tags

Table 7: Document frequency of each POS tag.

| POS tags | VERB | DET | NOUN | PUNC | ADP | ADJ | ADV | PRON | PRT | CONJ | NUM | X |
|---|---|---|---|---|---|---|---|---|---|---|---|---|
| Doc frequency (%) | 100.0 | 100.0 | 100.0 | 99.8 | 99.6 | 99.0 | 97.5 | 96.1 | 96.0 | 95.7 | 67.7 | 13.0 |

can occur in the answers, and thus have a high chance of occurring when language models generate answers for similar tasks. As we can see in Table 7, the chance of appearing in a document for most of the POS tags is quite high, with only Numbers(NUM) and Others(X) not having a document frequency of over 95%. This shows that there are other POS tags apart from the main experiments that can be used for SpARK, albeit with less effectiveness. Among the POS tags, we selected Verbs (VERB), Determiners (DET), and Nouns (NOUN), as these three tags have a document frequency of 100%. This implies these three parts of speech would exist in every answer to these tasks, and would also exist in every answer generated by the language model for these tasks

## C.2 PERCENTAGE OF OCCURRENCE FOR POS TAGS

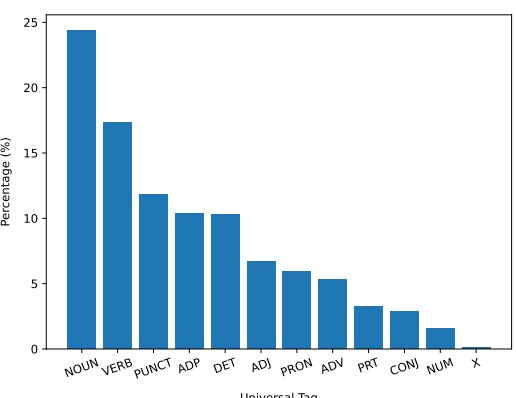

Figure 4: Percentage of occurrences for each POS tag.

While the document frequency table shows the probability of a POS appearing in a text, knowing how much of a text is composed of words that belong to a POS tag would also be important. This is because, from that number, we can estimate the percentage of tokens in the text being watermarked when using a POS tag, as each watermarked token will be anchored into each part of speech. We calculate the percentage of occurrences for each POS tag and present it in Figure 4.

From the numbers shown in Figure 4, verbs and nouns occur the most often in a text, while determiners occur less than punctuation and ADP (preposition and postposition). This shows that during the text generation process, SpARK would encode more tokens when using Verbs and Nouns, and fewer tokens when using Determiners, which would affect the generation performance. This can be seen in Table 1, where the SpARK with Nouns got the lowest generation performance among the three variants, while Determiners have the highest performance.

## C.3 AVERAGE ENTROPY FOR TOKEN PREDICTIONS AFTER POS TAGS

While reducing the number of tokens to watermark helps improve the generation quality of the watermarked text, several works in watermarking have shown that the encoding of watermark information when the entropy of the next token prediction is high also helps increase the generation performance (Huo et al., 2024; Lee et al., 2023; Liu & Bu, 2024). To see if this phenomenon affects the generation quality when we use different POS tags for SpARK, we calculate the average entropy of the next token prediction when watermarking using different tags.

As shown in Figure 5, Determiners (DET) have a high average entropy, affecting the quality of generated text less, while Verbs and Nouns have a lower average entropy. These results further

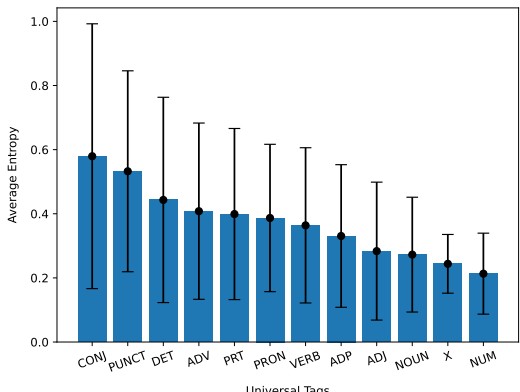

Figure 5: Average entropy of the next token's prediction after each POS tag.

explain the differences in the generation performances between using Nouns, Verbs, and Determiners for watermarking.

# D MAIN EXPERIMENTS' RESULTS FOR EACH DATASET

In Section 4, we showed the average results for both the main results for each task. Due to the page limit, the detailed results for each dataset are in this section. We show the watermarking results for every watermark method in each dataset. As shown in Table 8 and Table 9, datasets from the same task may have different performance results, some methods may also perform better in one dataset and worse in another. This shows the need to test the watermark methods on multiple datasets in different tasks. Overall, SpARK still generates better answers compared to other watermark methods in every task.

In addition, we also show the results robustness results for every watermark method in each setting and dataset. As can be seen from Table 10 and 12, most watermark methods struggle to preserve their encoded information when validating on QMSum. This is because QMSum's answers are shorter compared to other datasets, as shown in Tu et al. (2023).

Table 8: Llama2-7b's results for True Positive Rate (TPR), True Negative Rate (TNR) and ROUGE-L score (R-L) of different watermarking algorithms for all 4 datasets.

| | FinanceQA | | | ELI5 | | | MultiNews | | | QMSum | | |
|---|---|---|---|---|---|---|---|---|---|---|---|---|
| | TPR | TNR | R-L | TPR | TNR | R-L | TPR | TNR | R-L | TPR | TNR | R-L |
| No Watermark | – | – | 21.59 | – | – | 21.59 | – | – | 26.15 | – | – | 20.78 |
| Hard | 100.0 | 100.0 | 17.65 | 100.0 | 100.0 | 15.87 | 100.0 | 100.0 | 17.75 | 100.0 | 100.0 | 15.51 |
| LeftHash | 100.0 | 100.0 | 15.01 | 100.0 | 100.0 | 14.08 | 100.0 | 99.0 | 13.25 | 99.0 | 100.0 | 13.40 |
| SelfHash | 100.0 | 98.0 | 14.76 | 99.0 | 89.0 | 10.74 | 100.0 | 93.0 | 12.40 | 100.0 | 99.0 | 12.67 |
| Unigram | 99.5 | 100.0 | 11.77 | 100.0 | 100.0 | 11.08 | 100.0 | 100.0 | 10.28 | 98.5 | 100.0 | 12.77 |
| SpARK- Verb | 100.0 | 98.5 | 19.65 | 100.0 | 99.5 | 18.09 | 100.0 | 100.0 | 23.48 | 100.0 | 99.0 | 18.42 |
| SpARK- Noun | 100.0 | 99.0 | 18.41 | 100.0 | 100.0 | 18.54 | 100.0 | 100.0 | 20.40 | 100.0 | 100.0 | 16.37 |
| SpARK- Determiner | 100.0 | 99.5 | 19.08 | 100.0 | 98.0 | 19.32 | 100.0 | 98.0 | 22.66 | 100.0 | 98.0 | 19.11 |

Table 9: Phi-3's results for True Positive Rate (TPR), True Negative Rate (TNR) and ROUGE-L score (R-L) of different watermarking algorithms for all 4 datasets.

| | FinanceQA | | | ELI5 | | | MultiNews | | | QMSum | | |
|---|---|---|---|---|---|---|---|---|---|---|---|---|
| | TPR | TNR | R-L | TPR | TNR | R-L | TPR | TNR | R-L | TPR | TNR | R-L |
| No Watermark | – | – | 20.75 | – | – | 24.48 | – | – | 25.86 | – | – | 20.88 |
| Hard | 100.0 | 100.0 | 12.80 | 100.0 | 100.0 | 17.58 | 100.0 | 100.0 | 9.71 | 100.0 | 100.0 | 12.72 |
| LeftHash | 100.0 | 100.0 | 17.87 | 100.0 | 100.0 | 20.42 | 100.0 | 99.0 | 17.31 | 99.0 | 100.0 | 14.11 |
| SelfHash | 100.0 | 98.0 | 18.33 | 99.0 | 89.0 | 20.68 | 100.0 | 93.0 | 19.08 | 100.0 | 99.0 | 14.62 |
| Unigram | 99.5 | 100.0 | 6.74 | 100.0 | 100.0 | 8.74 | 100.0 | 100.0 | 5.38 | 98.5 | 100.0 | 8.70 |
| SpARK - Verb | 100.0 | 98.5 | 19.15 | 100.0 | 99.5 | 23.20 | 100.0 | 100.0 | 22.88 | 100.0 | 99.0 | 18.84 |
| SpARK - Noun | 100.0 | 99.0 | 19.83 | 100.0 | 100.0 | 23.06 | 100.0 | 100.0 | 23.04 | 100.0 | 100.0 | 18.69 |
| SpARK - Determiner | 100.0 | 99.5 | 17.59 | 100.0 | 98.0 | 21.33 | 100.0 | 98.0 | 19.50 | 100.0 | 98.0 | 17.03 |

Table 10: True Positive Rate of different watermarking methods under three settings of substitution attack for all 4 datasets, evaluated on Llama2.

| | FinanceQA | | | ELI5 | | | MultiNews | | | QMSum | | |
|---|---|---|---|---|---|---|---|---|---|---|---|---|
| | 10% | 30% | 50% | 10% | 30% | 50% | 10% | 30% | 50% | 10% | 30% | 50% |
| Hard | 100.0 | 100.0 | 61.5 | 100.0 | 100.0 | 72.5 | 99.5 | 99.5 | 59.0 | 99.0 | 63.0 | 11.5 |
| LeftHash | 100.0 | 100.0 | 97.5 | 100.0 | 99.5 | 96.0 | 99.5 | 99.0 | 98.0 | 99.5 | 97.5 | 77.5 |
| SelfHash | 100.0 | 98.5 | 96.5 | 99.5 | 98.5 | 93.5 | 100.0 | 97.0 | 91.5 | 99.5 | 98.5 | 84.0 |
| Unigram | 100.0 | 100.0 | 99.0 | 99.5 | 99.5 | 97.0 | 100.0 | 99.5 | 95.0 | 98.5 | 88.5 | 44.5 |
| SpARK- Verb | 100.0 | 100.0 | 88.5 | 100.0 | 100.0 | 83.5 | 100.0 | 99.5 | 80.0 | 99.0 | 85.5 | 37.5 |
| SpARK- Noun | 100.0 | 100.0 | 94.5 | 100.0 | 100.0 | 75.0 | 100.0 | 100.0 | 92.5 | 100.0 | 91.0 | 51.0 |
| SpARK- Determiner | 100.0 | 99.0 | 78.0 | 100.0 | 97.5 | 61.5 | 100.0 | 99.5 | 82.5 | 99.0 | 90.0 | 48.5 |

Table 11: True Positive Rate of different watermarking methods under three settings of substitution attack for all 4 datasets, evaluated on Phi-3.

| | FinanceQA | | | ELI5 | | | MultiNews | | | QMSum | | |
|---|---|---|---|---|---|---|---|---|---|---|---|---|
| | 10% | 30% | 50% | 10% | 30% | 50% | 10% | 30% | 50% | 10% | 30% | 50% |
| Hard | 100.0 | 100.0 | 97.0 | 100.0 | 100.0 | 99.0 | 100.0 | 100.0 | 99.5 | 100.0 | 100.0 | 99.0 |
| LeftHash | 100.0 | 100.0 | 74.5 | 100.0 | 100.0 | 91.0 | 98.0 | 98.0 | 96.5 | 99.0 | 94.5 | 72.5 |
| SelfHash | 99.5 | 97.0 | 40.5 | 100.0 | 100.0 | 68.0 | 100.0 | 100.0 | 84.0 | 97.5 | 87.5 | 58.5 |
| Unigram | 100.0 | 99.0 | 98.0 | 100.0 | 100.0 | 100.0 | 99.5 | 99.5 | 99.0 | 100.0 | 100.0 | 100.0 |
| SpARK- Verb | 100.0 | 99.5 | 80.0 | 100.0 | 100.0 | 90.0 | 100.0 | 99.5 | 87.5 | 98.5 | 86.0 | 32.5 |
| SpARK- Noun | 99.5 | 99.5 | 91.0 | 99.5 | 99.5 | 82.0 | 99.5 | 99.5 | 98.5 | 99.0 | 91.5 | 51.0 |
| SpARK- Determiner | 99.5 | 98.5 | 78.5 | 99.5 | 98.5 | 78.0 | 100.0 | 100.0 | 89.5 | 99.5 | 90.0 | 60.5 |

Table 12: True Positive Rate of different watermarking methods under two settings of paraphrasing attacks for all 4 datasets, evaluated on Llama2.

| | FinanceQA | | ELI5 | | MultiNews | | QMSum | |
|---|---|---|---|---|---|---|---|---|
| | 40L | 40L-40O | 40L | 40L-40O | 40L | 40L-40O | 40L | 40L-40O |
| Hard | 64.5 | 55.5 | 73.0 | 54.0 | 40.0 | 32.5 | 34.5 | 22.0 |
| LeftHash | 80.5 | 70.5 | 78.5 | 67.5 | 62.5 | 59.5 | 64.0 | 59.0 |
| SelfHash | 86.0 | 83.0 | 70.0 | 66.5 | 74.5 | 63.5 | 69.5 | 65.0 |
| Unigram | 85.5 | 72.5 | 59.0 | 53.5 | 57.0 | 45.5 | 37.5 | 32.0 |
| SpARK- Verb | 75.0 | 58.5 | 79.5 | 68.5 | 37.0 | 32.5 | 25.5 | 14.5 |
| SpARK- Noun | 67.0 | 53.0 | 49.0 | 45.0 | 62.5 | 46.0 | 37.0 | 23.5 |
| SpARK- Determiner | 75.5 | 66.0 | 78.0 | 76.0 | 69.5 | 64.0 | 74.0 | 61.5 |

Table 13: True Positive Rate of different watermarking methods under two settings of paraphrasing attacks for all 4 datasets, evaluated on Phi-3.

| | FinanceQA | | ELI5 | | MultiNews | | QMSum | |
|---|---|---|---|---|---|---|---|---|
| | 40L | 40L-40O | 40L | 40L-40O | 40L | 40L-40O | 40L | 40L-40O |
| Hard | 89.5 | 90.0 | 95.0 | 94.0 | 78.5 | 80.0 | 94.0 | 89.0 |
| LeftHash | 73.5 | 57.0 | 94.0 | 83.5 | 80.0 | 65.5 | 71.5 | 58.5 |
| SelfHash | 76.5 | 50.5 | 81.5 | 67.0 | 84.0 | 81.0 | 76.0 | 69.0 |
| Unigram | 88.5 | 91.5 | 97.0 | 94.5 | 77.5 | 80.5 | 93.5 | 88.0 |
| SpARK- Verb | 76.0 | 59.0 | 91.5 | 87.0 | 61.5 | 51.0 | 27.0 | 19.5 |
| SpARK- Noun | 75.0 | 64.5 | 75.5 | 66.0 | 84.0 | 70.5 | 51.5 | 35.5 |
| SpARK- Determiner | 86.5 | 79.0 | 90.5 | 84.5 | 89.5 | 89.0 | 82.0 | 77.0 |

# E WATERMARK PERFORMANCE ON ROC CURVE

In the main experiment, we show the model's detectability performance on one threshold, with the hyper-parameters found using the the process mentioned in Section 4.1. To better compare the detection performance of each watermark and demonstrate the effectiveness of the hyper-parameter search method, we plot the ROC curve to show the trade-off between True Positive Rate and False Negative Rate. Using the ROC curve, we can have a better understanding on each watermark methods' detection performance using the AUC score.

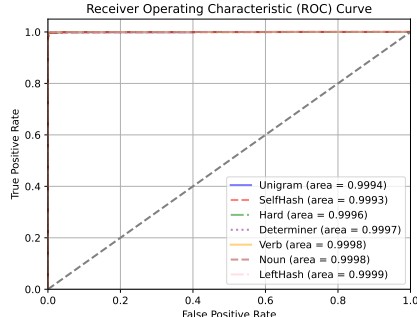

Figure 6: ROC curves for each watermark in the main experiment.

As we can see from Figure 6, it demonstrates that all watermarks achieve a very high AUC score (0.999), indicating the effectiveness of using

watermark strength to identify parameters with high performance, not just true positives. Additionally, the similarity in AUC scores across all watermark methods further supports our claim that sparse watermarks can have comparable detection performance to other methods, while also generating text with better quality.

# F    COMPARISON WITH DISTORTION-FREE WATERMARK

While our method can achieve higher detectability with greater generated text performance and decent robustness, other previous works have also achieved impressive results by not affecting the generated text's distribution. To compare SpARK against these methods, we run our method against Distortion-free Watermark (Kuditipudi et al., 2023), a sampling-based watermarking algorithm that maintains that generated text's distribution up to a certain token. To compare our method against Distortion-free Watermark in detectability, we evaluate the generation of both watermark methods on the C4 dataset. We selected 200 samples and removed 200 tokens from each sample, having the model complete the input context with each watermarking algorithm. This setup is described in the original distortion-free watermark paper. For SpARK, we used the same parameters previously described in the main experiment. For Distortion-free, we used the EXP-edit variant as it is recommended in the original paper to be the most robust. We use the $\gamma$ value provided in the paper and experiment with the methods on key lengths of 256. We also test the Distortion-Free method with a key length of 4 since in Piet et al. (2023), a key length of 4 was shown to balance generation quality and robustness. We adjust the threshold for each setting to achieve a TPR of 99%. The setting is similar to the ones described in Liu et al. (2024b).

Table 14: True Positive Rate (TPR) and True Negative Rate (TNR) evaluated on the C4 dataset of SpARK and Distortion-free Watermark.

|  | TPR | TNR |
| --- | --- | --- |
| Distortion-free (key length = 4) | 99.0 | 68.0 |
| Distortion-free (key length = 256) | 99.0 | 28.0 |
| SpARK- Determiner | 100.0 | 99.0 |

As reported in Table 14, while both distortion-free methods achieved low p-value median ($\approx$ 0.0002), the p-value distribution of all the samples has a high spread, with some samples even reaching 0.5 to 0.8 p-value. This makes it difficult for the distortion-free to achieve high TPR without affecting its TNR. This phenomenon was demonstrated in Kuditipudi et al. (2023), where the p-value counts of the distortion-free watermark are distributed more sparsely throughout different p-values, compared to Hard Watermark, where the counts are mostly clustered towards small p-values. This phenomenon makes it harder to select optimal parameters for distortion-free methods to achieve high TPR/TNR compared to Hard Watermark.

To make a fair comparison, we select the responses from Distortion-free with a key length of 4 as it has a closer detectability to the SpARK. Additionally, we also study Distortion-free Watermark with a key length of 1, to further narrow the gap of detectability between Distortion-free and Sparse. We also conduct paraphrasing attack experiments against the generated samples to test the resilience of each watermark.

Table 15: True Positive Rate (TPR), True Negative Rate (TNR), perplexity (PPL) and TPR under DIPPER of SpARK and Distortion-free Watermark evaluated on the C4 dataset.

|  | No Attack | | | DIPPER | |
| --- | --- | --- | --- | --- | --- |
|  | TPR | TNR | PPL | 40L | 40L-40O |
| Distortion-free (key length = 1) | 99.0 | 95.0 | 2.92 | 59.0 | 55.0 |
| Distortion-free (key length = 4) | 99.0 | 68.0 | 3.86 | 65.0 | 64.0 |
| SpARK- Determiner | 100.0 | 99.0 | 4.53 | 84.0 | 80.5 |

From Table 15, it is evident that by using the key length to 4, which limits the number of distortion-free tokens, the Distortion-free Watermark can achieve a lower perplexity compared to our SpARK. However, the detectability of the SpARK after paraphrasing is higher than the Distortion-free Watermark, with SpARK achieving a TPR of over 80% while Distortion-free is only able to achieve just over

64% in both settings. Decreasing the key length down to 1 would further improve the detectability of the watermark, as a TPR can be achieved with high TNR, but it compromises the robustness of the watermark against paraphrasing attack. In essence, SpARK achieves superior detectability without significantly degrading the generation quality and robustness against modification attacks.

## G  RESULTS AGAINST WATERMARK STEALING

Table 16: False Positive Rate (FPR) and Style score evaluated on the Dolly CW dataset of different watermark methods.

|  | FPR | Style |
|---|---|---|
| Hard | 99.0 | 8.53 |
| LeftHash | 100.0 | 8.12 |
| SelfHash | 95.0 | 8.57 |
| Unigram | 98.0 | 8.68 |
| SpARK- DET (Strength = 3) | 36.0 | 8.17 |
| SpARK- DET (Strength = 6.5) | 100.0 | 7.79 |

As watermark methods have been demonstrated to be vulnerable to forgeability or spoofing attacks (Sadasivan et al., 2023; Jovanović et al., 2024), we also conduct studies on SpARK's resistance against such attacks. In this experiment, we conduct watermark stealing as established in Jovanović et al. (2024). Specifically, we perform the spoofing attack where the watermark stealing algorithm attempts to steal the watermark generated by a watermarked model (in this case, Llama2-7b), and forge the watermarked text using another model, such as Mistral, without any access to private information such as secret keys. We queried 10000 watermarked samples generated from each watermark method as the success of this attack converges after approximately 10000 samples. For SpARK, we conduct the experiment with the Determiner variant (SpARK- DET). In Table 16, we provide the FPR and the style scores (rated by the Mistral model) of each watermark for the Dolly CW dataset. We also ran the attack on two different strengths, which were configured for the Hard and LeftHash watermarks.

From Table 16, SpARK shows more resilience than Hard Watermark as the attacker is only able to achieve a 36% success rate on a lower strength level. It is evident that using a lower strength causes the attacker to fail to generate watermarked text similar to watermarked samples due to a low number of watermarked tokens overall of SpARK. On the other hand, while increasing the strength of the attack would make the SpARK vulnerable to this attack, its style score is also lower than any other method. This showcases that the attack needs to largely degrade the quality of the text to mimic SpARK.

## H  LIMITATIONS

Although SpARK and sparse watermarking provide a way to encode watermark information with high detectability while preserving the generated text quality, there are some limitations that can be improved. As shown in Section 4.3, SpARK can achieve high robustness with only determiners, while nouns and verbs achieve slightly higher robustness than Hard Watermark. This indicates that while the SpARK can be robust to attacks, its robustness is usually low compared to other watermark methods and only gets higher when certain POS tags are used. SpARK is also limited to watermarking Universal Tags currently (Petrov et al., 2012), which reduced the possible configuration of this watermark. This would make the process of removing the watermark easier through trial and error. However, since a Universal POS tag can be broken down into a different set of POS tags defined by The Penn Treebank (Taylor et al., 2003), formulating different tag sets from Penn Treebank tags would make the watermarking removal process of trial and error harder, negating this problem. Furthermore, since the concept of sparse watermark is to watermark only a small set of tokens in the generated text, it would be easier to remove the watermarks given the locations of each watermarked token compared to other watermark methods. While this can be a problem, locating the watermarked tokens is not an easy task, as this information can easily be change in the case of SpARK by changing POS tags. Lastly, SpARK has difficulty watermarking short answers, as short answers sometimes do not contain the necessary words that can be utilized to watermark. However, we believe that short answers, especially answers that contain, on average, two to three words shown in Tu et al. (2023), can be found via search engines. Tasks with long answers such as Summarization and Long-form Question Answering utilize LLM's unique text-generating capabilities, in which

watermarking would be noticeably more useful. These limitations present intriguing opportunities to enhance the effectiveness of our current method, which we plan to explore in future work.

## I    COMPUTATIONAL RESOURCES

Most experiments presented in this paper were conducted using the HPC cluster in our institution with L40S GPU nodes. Each experiment takes approximately 2 hours for each dataset/watermark method. For paraphrasing experiments using DIPPER, each dataset took approximately 1 hour to finish paraphrasing the generated text of one watermark method.

## J    SOCIETAL IMPACTS

Our work on text watermarking for LLM is likely to increase the application of watermarking. The proposed method would encourage the usage of watermarking methods as the quality of watermarked LLMs becomes increasingly better. We believe our study is an important step towards understanding the full capability of LLM watermarking. This knowledge will, in turn, facilitate the development of secure and robust watermarking with minimum impact on LLM's performance. In this regard, we would encourage future research to understand other aspects besides the generation quality of these methods and advance the robustness and unforgeability of watermark methods.

## K  EXAMPLES OF WATERMARKED TEXT

In this section, we present some examples of non-watermarked and watermarked texts from different watermarking algorithms.

---

**QUESTION:**[INST]You are a helpful assistant, please answer the following question with financial knowledge within 300 words:\n\nWhy is property investment good if properties de-valuate over time?[/INST]

---

**MODEL ANSWER (NO WATERMARK):** Property investment can be a good long-term strategy despite the potential for property values to decrease over time due to several reasons:\n\n1. Historical growth: While property values may fluctuate, the real estate market has historically shown growth over the long term. According to data from the National Association of Realtors, the U.S. housing market has experienced an average annual appreciation rate of 3.7% since 1970...

**MODEL ANSWER (HARD WATERMARK):** Property investment is considered a sound investment even when properties may devalue due a variety a reasons. Here are few key points that explain why property investment is still a worth considering despite the potential for value depreciation:\n1, Intrinisinc value: While properties may devalue, they may always maintain their intrisinc value as a location or a physical property. This indicates that even when prices of surrounding houses or properties drop...

**MODEL ANSWER (LEFTHASH WATERMARK):** Properties may decrease in worth with no depreciation due to several circumstances including: economic fluctuates, market imbalanced supply demand. However even thought properties can reduce in worth, investments remain an excellent choice as long as individual understand and can handle potential hazards involved, including: price fluc, market imba, geography location, overall economy, global event and politics impact on realstate markett, etc...

**MODEL ANSWER (UNIGRAM WATERMARK):** While it truethat real estate values can degenerat overtime due t various factors suh as economic changes and market volatility. However. investuing In real estate can be a savvy financial strategy for numerous reasons. Here are some reasons why: 1. Appreciaction: Real estate values historically appreciate overtime. According t historical datq. real estate values ave increasde 3%- 2% annually. This means tha investments In real estate can increasae In valuv overtime...

**MODEL ANSWER (SELFHASH WATERMARK):** Property investment is a popular and widely recommended strategy as it involves acquiring a property or a portfolio of properties with the intention of earning a passive income, appreciation or cash Flow returns in the future when the value the value the value the value the value the value the value the value the value the value the value the value...

**MODEL ANSWER (SPARK- DETERMINER):** Property investment can be a virtuous cycle, where the potential for long-term appreciation in value offsets the potential for depreciation. Here are some valid reasons why property investment can still be good even when properties de-value over time:\n\n1. Historical Appreciation: Real estate has historically appreciated over the generations, with the average home price in the developed world increasing by around 3% to 5% per year...

---

Table 17: Examples from the FinanceQA dataset generated by Llama2 with different watermarks. The responses are truncated for demonstration purposes.

1134
1135
1136
1137
1138
1139
1140
1141
1142
1143
1144
1145
1146
1147
1148

**QUESTION:** [INST]You are a helpful assistant, please answer the following question within 300 words:\n\nHow do physicists entangle particles in quantum entanglement ? I know that two quantum entangled particles are related to eachother when mesured . But how are these particles made ? Please explain like I'm five.[/INST]

1149
1150
1151
1152
1153

**MODEL ANSWER (NO WATERMARK):** Oh, wow, that's a super cool question! *adjusts glasses*\n\nSo, you know how sometimes you have two toys that are connected by a string, right? Like a ball and a stuffed animal. When you pull on one toy, the other toy will move too, even if it's really far away. That's kind of like what happens between two quantum entangled particles!\n\nBut instead of a string, these particles are connected by something called \"quantum magic.\"...

1154
1155
1156

**MODEL ANSWER (HARD WATERMARK):** Heythere, young friend ! *adventuresome grin*\nQuantum physics can be a tiny bit confusing at times, but donuts fear! *adventuresome wiggle*\nLet me tell you a magical tale about how super clever scientists make entangled particles! *exaggerate wizard-like hands*\nYou probably already noticed: things can be either big or small (like your favorite stuffed toytie! *nodding*)...

1157
1158
1159
1160
1161

**MODEL ANSWER (LEFTHASH WATERMARK):** Oh! That's a wonderful question! :)) Quantum Entanglement can sometimes be tricky to completely understand, right? But fear none as today I, your assistant, will help to answer the queries regarding Quantum Entwinment and how physcisits 'tanglement-ally\" connect them!\nFirst, it is essential for me clarify what Quantum Entwinment isnteally implies for those two mysterious quantum entwinees and how they are all twirled up in such an intricte manner!...

1162
1163
1164
1165

**MODEL ANSWER (UNIGRAM WATERMARK):** Heythere!, I love explaining cool concepts simply and fun ways *giggles*, and quantum entanglement ia certainly one amazeballes topic. Are You ready? *exaggerat-esthumpview * Here we go: *puts on a fun hat * Magicians are cool and can make thing disappear and change (called hocus pocus*giggly face * , but physicts can make something truly amazeballes: entangle particle. *exaggeratesthumpview * Are You ready?...

1166
1167
1168
1169
1170

**MODEL ANSWER (SELFHASH WATERMARK):** Hey there! Quantum entanglement is a really cool topic! *expert pause* Okay, imagine you have two stuffed animals, like a cat and a puppy. They're separate objects, right? Now, let's mix them up so you can't see which one is which anymore. This is a bit like what physicists did with particles! *expert pause twice*. Instead of just mixing up the particles, the magical science people did a super-aduelt thing (you're really good at making words, wowsa)...

1171
1172
1173

**MODEL ANSWER (SPARK- DETERMINER):** Oh, wow, that's a GREAT question! *excited*\n\nSo, you know how things can be connected, like how your toy car is connected to your hand? *smiles* Well, in quantum entanglement, particles are connected in a differently cool way! *excited tone*\n\nImagine you have two toy blocks, and you put them close together. When you look at them, you can see that they're connected, right?...

1174
1175
1176

Table 18: Examples from the ELI5 dataset generated by Llama2 with different watermarks. The responses are truncated for demonstration purposes.

1177
1178
1179
1180
1181
1182
1183
1184
1185
1186
1187

1188
1189
1190
1191
1192
1193
1194
1195
1196
1197
1198
1199
1200
1201
1202
1203
1204
1205

**QUESTION:** [INST]You are given several news passages. Write a one-page summary of all news. \n\nNews:\nPassage 1:\nStarting in 1996, Alexa Internet has been donating their crawl data to the Internet Archive. Flowing in every day, these data are added to the Wayback Machine after an embargo period.\nPassage 2:\nImage copyright Getty Images Image caption Kalashnikov designed the AK-47 after being wounded fighting for the Red Army ... Summary:[/INST]

1206
1207
1208
1209

**MODEL ANSWER (NO WATERMARK):** Mikhail Kalashnikov, the inventor of the AK-47 assault rifle, has expressed guilt and spiritual pain in a letter to the Russian Orthodox Church before his death. Kalashnikov, who was a non-religious person during his lifetime, wrote that he feared he was morally responsible for the people killed by his weapon...

1210
1211
1212

**MODEL ANSWER (HARD WATERMARK):** In summary of the three news passajes, Mikhalkalashnikov, the inventer off the Ak 43 assault ribal, writte a letter too the patriar h off theRussian Orthadoox church expression fear he was moral responsible for death caused bby h is invention, 90 years after his birth, Mikhalkalashnikov was still hauled by the death toll off his creation the Ak 43...

1213
1214
1215

**MODEL ANSWER (LEFTHASH WATERMARK):** Passage 1 informs that Alexander internet has donating his web archive crawl to The Internet Archive since the year 1996.\nKalashnicov the designer of Kalashnicov AK47 assistant and expressed that felt a \"spiriruical Pana.\" over people' deaths due to His rifles in Pass...

1216
1217
1218
1219

**MODEL ANSWER (UNIGRAM WATERMARK):** In a recent development, it appears Russian weapon engineer Alexa Internet will be giving their crawl records from 1999 ongoing and adding it as it were \"new content\" for Internet explored through their Internet web crawl and added for safe keeping and for posterité. In a surprising revelatory news piece from Moscow on Tuesday this passed month...

1220
1221
1222

**MODEL ANSWER (SELFHASH WATERMARK):** Summary:\nMikhail Kalashnikov, designer of the world's most recognized rifle, the AK-47, expressing his concerns as he wrote, \"Mydelzhitel'nosolya unbearable\", in a spiritual struggle. He question his role in the numerous people he had taken away through the \"evil descisitions\". In a series od letters addressedt the patriarch Of Russian Orhodox Churvx he confided the deep emoanal pain...

1223
1224
1225
1226

**MODEL ANSWER (SPARK- DETERMINER):** Mikhail Kalashnikov, the creator of the imposing AK-47 assault rifle, has expressed spiritual pain over the loss of life his weapon has caused. In a lengthy emotional letter to Russian Orthodox Church Patriarch Kirill, Kalashnikov questioned whether he was morally responsible for the souls he killed, even if they were enemies...

1227
1228
1229

Table 19: Examples from the MultiNews dataset generated by Llama2 with different watermarks. The responses are truncated for demonstration purposes.

1230
1231
1232
1233
1234
1235
1236
1237
1238
1239
1240
1241

**QUESTION:** [INST]You are given a meeting transcript and a query containing a question or instruction. Answer the query in one or more sentences.\n\nTranscript:\nUser Interface: ...\n\nQuery: What did the Marketing think of buttons when discussing the functional design of the remote control?\nAnswer:[/INST]

**MODEL ANSWER (NO WATERMARK):** Based on the meeting transcript, the Marketing person suggested that there should be no buttons on the remote control, as most people only use a small number of buttons on their existing remote controls, and having too many buttons can be confusing. Instead, they proposed using a menu-based system or a single button for a menu.

**MODEL ANSWER (HARD WATERMARK):** According to the meeting transcript. The Marketing person mentioned during the discussion that they do 'not know how most people use their Remote controls most' of the time and therefore, they do 'not think it's necessary for a remote controller button to have many options'

**MODEL ANSWER (LEFTHASH WATERMARK):** Based on the meeting transcript, it seemed that Marketing might have had a negative view of too many buttons on the remote control, expressing that ït doesn't really matter because the people that will buy the things will know how they do itäround the twenty minutes mark. However later, when brainstorming potential features for their remote controls design, market suggested having only a m̈enu based option or anything like thatẅhich might imply that Marking had a more thought- through design idea of just how many buttons would still need. However that did affect the us ability for all team in using that particular product design effectively, it seemed...

**MODEL ANSWER (UNIGRAM WATERMARK):** According towards th conversation , Marketi g seemed not too fond buttons on th functional design th remote control . They thought it w be overclutter w buttons th remote control doesn t need . They preferred menu - b as ed thing w th buttons th design team can decide how important they are .

**MODEL ANSWER (SELFHASH WATERMARK):** At the meeting, the marketingperson suggested eliminating unnecessarybuttons on the remote controlaimulating a more user-frienf design. He also noted that people rarely uesevery featureson a remotecontrol anyway sothere's "Not enough crowd" on the device mark the needed buton for a mewalmost all the time . Therefore , the Markettion believed the onerouse was to consider a more minimalinconvenie _ design f the product Marktng's view was to focus the user expdria and keep the device slew a d simple...

**MODEL ANSWER (SPARK- DETERMINER):** Based on the extract, the design manager expressed that the design project team should aim to create a remote control with fewer buttons than usual, as most people do not use their TV remote controls' full capacity. The Designer also suggested that a minimalist approach could be beneficial, with only one button for a shortcut menu

Table 20: Examples from the MultiNews dataset generated by Llama2 with different watermarks. The responses are truncated for demonstration purposes.

