# OpenReview forum: "Sparse Watermarking in LLMs with Enhanced Text Quality"
_ICLR.cc/2025/Conference — Submitted to ICLR 2025_

### Official Review · Reviewer_FdiC · 2024-10-31

**Soundness:** 2
**Presentation:** 3
**Contribution:** 2
**Rating:** 3
**Confidence:** 4

**Summary:**

The paper introduces a watermarking technique called SpARK for Large Language Models (LLMs), which selectively embeds watermarks in a small subset of generated tokens. This approach aims to maintain high text quality while ensuring watermark detectability. Experimental results demonstrate that SpARK outperforms traditional watermarking methods in text quality across various tasks and effectively withstands common attacks, such as substitution and paraphrasing. By adding watermarks to tokens of specific part-of-speech tags and only detecting these specific tokens, this method preserves text quality without compromising detectability.

**Strengths:**

1. Although the idea may seem simple, it effectively maintains text quality by focusing watermarking on tokens with specific part-of-speech (POS) tags. This strategy helps mitigate robustness issues caused by token insertion or substitution in other positions.
2. Compared to traditional methods such as Hard Red-Green, Soft Red-Green, LeftHash, SelfHash, and Unigram, SpARK demonstrates a clear advantage in preserving text quality during watermarking.
3. The paper provides a thorough analysis of robustness against various attacks and considers resistance to watermark stealing, showcasing the method's effectiveness in real-world scenarios.

**Weaknesses:**

1. The main consideration of the paper is to preserve text quality; however, the baseline methods compared in the main experiments are traditional and do not primarily focus on preserving text quality.
- While there is a comparison with distortion-free methods in Appendix F, the only measure for text quality is perplexity (PPL), which is insufficient, and SpARK does not demonstrate an advantage over distortion-free methods.
- Additionally, other methods that could aid in preserving text quality, such as unbiased watermark[1], DiPMark[2], TS-Watermark[3], and SWEET[4], were not included in the experiments.

Reference:

[1] Unbiased Watermark for Large Language Models.

[2] DiPMark: A Stealthy, Efficient and Resilient Watermark for Large Language Models.

[3] Token-Specific Watermarking with Enhanced Detectability and Semantic Coherence for Large Language Models.

[4] Who Wrote this Code? Watermarking for Code Generation.

**Questions:**

See the Weaknesses section.

---

> ### Author Response · Authors · 2024-11-22
> **Rebuttal by Authors**
>
> Thank you for the comments, please see our responses below.
>
> **Q1:The main consideration of the paper is to preserve text quality; however, the baseline methods compared in the main experiments are traditional and do not primarily focus on preserving text quality.**
>
> > **A:** Thank you for your suggestion. When conducting the experiment, our objective was to introduce and evaluate the Sparse Watermarking (SpARK) framework against widely used watermarking techniques to establish a broad understanding of its advantages. For comparison with methods that helped preserve text quality, we ran the EWD watermark on the WaterBench baseline, the results of which are presented in the Public Global comment. In short, we show that while EWD produces better quality answers compared to other watermark methods, **SpARK** overall still has a higher Rouge-L score.
>
>
> **Q2:While there is a comparison with distortion-free methods in Appendix F, the only measure for text quality is perplexity (PPL), which is insufficient, and SpARK does not demonstrate an advantage over distortion-free methods.**
>
> > **A:** Thank you for your comment. While Distortion Free was able to have a slightly lower perplexity than ours, our method is still robust to paraphrasing attacks. Furthermore, distortion-free cannot achieve as high of detectability compared to **SpARK**, as the result also shows that the shortest key (length = 1) with the lowest p-value median cannot achieve detectability similar to other methods. Meanwhile, **SpARK** is more flexible, and can easily change the trade-off between text quality and detectability by changing gamma. Lastly, Distortion Free takes significantly more time to detect its watermark compared to ours. From the table below, we can see that the generation time for each of the sample text is not so different between the two methods. However, the time it takes to detect a watermark per sample for Distortion Free watermark is a lot more compared to **SpARK**. This shows that our method is more suitable for real-world application compared to Distortion Free watermark.
> >
>
> |                 | Generation Time (s) | Detection Time (s) |
> |-----------------|---------------------|--------------------|
> | Distortion Free | 11.49               | 101.59             |
> | **SpARK**       | 12.98               | 1.11               |
>
> **Q3:Additionally, other methods that could aid in preserving text quality, such as unbiased watermark, DiPMark, TS-Watermark, and SWEET, were not included in the experiments.**
>
>
> > **A:** Thank you for your suggestions. As stated previously, our objective was to introduce and evaluate the Sparse Watermarking (SpARK) framework against widely used watermarking techniques to establish a broad understanding of its advantages. To further improve our claims of **SpARK**'s effectiveness, we have also run a comparison between our method and other quality-preserving text watermarks which are EWD[1] and SWEET[2]. The experiment results are presented in the Public Global comment, but to summarize, **SpARK** was still able to maintain higher quality compared to EWD and SWEET, while maintaining detectability after paraphrasing attacks. This shows that **SpARK**'s watermarked text struck a balance between high-quality generations and robustness to attacks.
>
>
> [1] Lu et al., An Entropy-based Text Watermarking Detection Method. 2024.
>
> [2] Lee et al., Who Wrote this Code? Watermarking for Code Generation. ACL 2024

---

> > ### Comment · Reviewer_FdiC · 2024-12-03
> >
> > Thank you for your response and the additional experiments. However, I find the design of the sparse watermark to be somewhat trivial. It's not that simplicity is inherently bad, but I do not see a theoretical advantage over existing sparse watermarks like SWEET or EWD. Additionally, the experimental advantages are not particularly evident. Therefore, I will maintain my score.

---

> ### Author Response · Authors · 2024-12-03
> **Response from the Author**
>
> Thank you for the comments, please see our responses below.
>
> **Q4: Thank you for your response and the additional experiments. However, I find the design of the sparse watermark to be somewhat trivial. It's not that simplicity is inherently bad, but I do not see a theoretical advantage over existing sparse watermarks like SWEET or EWD. Additionally, the experimental advantages are not particularly evident. Therefore, I will maintain my score.**
>
> > **A:** Thank you for your response. While the theory of sparse watermark has been presented in  SWEET (note that EWD is not sparse watermarking), SWEET only focuses on low entropy tasks (i.e., code generation); our work addresses two major limitations inherent in SWEET. Firstly, our method does not require access to the original prompts associated with each text, nor does it depend on the original model to detect watermarks; the first requirement (access to the original prompts) only works for code generation, but not in the general context of text generation, while the second requirement significantly increases the computational cost of detecting a watermarked sample. **SpARK** does not have either of these requirements (or restrictions). Secondly, **SpARK**  was able to achieve slightly better empirical performance, while being more robust under paraphrasing, compared to SWEET (and also EWD, although EWD is not sparse watermarking) without the restrictions we just mentioned; **SpARK** achieved a better Rouge-L score compared to SWEET (and also EWD), and had significantly better results in Summarization (reducing 10% less of the original Rouge-L score), while maintaining slightly better results on LFQA  (1.5% less Rouge-L score reduction). For these reasons, we believe that **SpARK** has unique advantages and contributions compared to the previous works.
>
> > We also believe that the "simplicity" of **SpARK** (which the Reviewer also appreciates) is actually its major advantage, as it enables **SpARK** to be **practical**, while being able to perform extremely well compared to other watermarking methods (including the prior Sparse Watermarking ones) and overcome major disadvantages of previous Sparse Watermarking works. Designing such a simple, practical, and high-performing watermarking method is not a trivial task and we sincerely hope that our clarifications in this response can convince the Reviewer to change the rating of the work.

---

### Official Review · Reviewer_KCSa · 2024-11-02

**Soundness:** 2
**Presentation:** 2
**Contribution:** 1
**Rating:** 3
**Confidence:** 4

**Summary:**

The authors present "SpARK," a sparse watermarking method for large language models (LLMs), claiming that it achieves high detectability with minimal impact on text quality by watermarking a subset of generated tokens linked to specific POS tags. While the concept is well-explored, the novelty of this approach is questionable.

**Strengths:**

1. Innovative Use of POS Tags: The paper introduces an innovative approach by associating watermarks with specific Part-of-Speech (POS) tags, allowing for selective, sparse embedding. This method cleverly leverages the natural structure of language, enhancing the watermark's subtlety and detectability.

2. High Detection Performance: The experiments demonstrate that SpARK achieves near-state-of-the-art detection performance (e.g., True Positive Rate and True Negative Rate close to 100%) across various datasets, while maintaining minimal text quality degradation.

**Weaknesses:**

1. Lack of Innovation: The core idea of applying watermarks to a sparse subset of generated tokens is not novel. Existing works have already explored similar sparse watermarking and selective token modification methods. The paper fails to demonstrate a significant deviation from or advancement beyond these existing solutions. Recent watermarking studies already emphasize limiting the impact on text quality while ensuring robustness, making SpARK's claims less compelling.

2. Questionable Benefit Claims: The authors argue that SpARK's sparse watermarking reduces the degradation of text quality compared to dense watermarking. However, numerous recent works on text watermarking already achieve minimal or no impact on text quality. These techniques do not require sparsity to maintain text integrity, suggesting that the claimed advantage of SpARK may not be as impactful as presented.

3. Evaluation Limitations: While the authors provide experimental results demonstrating the performance of SpARK, the evaluations do not sufficiently differentiate it from existing sparse watermarking techniques. The lack of comparative analysis with newer and more diverse watermarking approaches that similarly preserve text quality undermines the validity of the claimed improvements.

Reference: Adaptive Text Watermark for Large Language Models

**Questions:**

1. What are the advantages of your method compared to distortion-free watermarking methods in LLMs?

2. The method proposed in the paper demonstrates a certain level of robustness against lexical substitution attacks. The paper notes that the method effectively maintains robustness when facing a 10% substitution rate, but as the substitution rate increases (e.g., 30% or 50%), the detection accuracy declines. My question is: Do you consider maintaining robustness against a 10% substitution rate as indicative of strong robustness? Additionally, is 10% sufficient for real-world application scenarios?

---

> ### Author Response · Authors · 2024-11-22
> **Rebuttal by Authors**
>
> Thank you for the comments, please see our responses below.
>
> **Q1:Lack of Innovation: The core idea of applying watermarks to a sparse subset of generated tokens is not novel. Existing works have already explored similar sparse watermarking and selective token modification methods. The paper fails to demonstrate a significant deviation from or advancement beyond these existing solutions. Recent watermarking studies already emphasize limiting the impact on text quality while ensuring robustness, making SpARK's claims less compelling**
>
> > **A:** Thank you for your question. As mentioned in our paper, our goal is to explore the concept of a sparse watermark. However, our concept of sparse watermark is not only to limit the number of tokens used for watermarking but also to limit the detection process to a small subset of tokens that is closely tied to the grammatical structure of the generated text. This will incorporate the benefits of higher-quality text generation during sparse watermarking, while maintaining the detectability and robustness of the embedded watermark.
> >
> > While there exist methods (i.e. SWEET) with similar concepts to ours, our novelty lies in sparse watermarking without any access to additional information such as the original prompts or the original model, which are the main limitations of the existing sparse watermarking methods.
>
> **Q2:Questionable Benefit Claims: The authors argue that SpARK's sparse watermarking reduces the degradation of text quality compared to dense watermarking. However, numerous recent works on text watermarking already achieve minimal or no impact on text quality. These techniques do not require sparsity to maintain text integrity, suggesting that the claimed advantage of SpARK may not be as impactful as presented.**
>
> >**A:** Thank you for your comment. In our paper, we have run an experiment comparing our method to  Distortion Free watermark, a watermark method that aims to minimize its impact on the text quality. We strongly believe that we have thoroughly discussed and compared our work to existing watermarks with similar goals of having minimal impact on text quality; nevertheless, we would greatly appreciate it if the reviewer could kindly point out any specific relevant work that may be missing from our paper. We are more than willing to promptly discuss and evaluate these works during the rebuttal process.
>
> **Q3:Evaluation Limitations: While the authors provide experimental results demonstrating the performance of SpARK, the evaluations do not sufficiently differentiate it from existing sparse watermarking techniques. The lack of comparative analysis with newer and more diverse watermarking approaches that similarly preserve text quality undermines the validity of the claimed improvements.**
>
> > **A:** Thank you for your suggestion. However, we believe that we have addressed the other sparse watermark methods in the related section and pointed out the differences between their methods and ours. We have also run additional experiments comparing our method to SWEET, a watermark method that sparsely watermarks code generation, a distinction to our work that we have acknowledged in our paper. The result is available in the Public Global Comment. Nevertheless, we would appreciate it if the reviewer points out any specific relevant work that may be missing from our paper. We are willing to discuss and evaluate those works during the rebuttal process

---

> > ### Author Response · Authors · 2024-11-22
> > **Continued the rebuttal comments.**
> >
> > **Q4:What are the advantages of your method compared to distortion-free watermarking methods in LLMs?**
> > > **A:** Our method offers several key advantages over distortion-free watermarking methods in large language models, as demonstrated in the results. For instance, as shown in Table 15 (Appendix Section F), our Sparse Watermark consistently outperforms distortion-free methods in terms of detectability, particularly in paraphrasing scenarios. Sparse Watermark achieves over 80% detectability, whereas distortion-free methods achieve just over 64% in comparable settings. This indicates that our method is more robust in maintaining watermark integrity, even when the text undergoes transformations like paraphrasing.
> > >
> > > A notable limitation of distortion-free watermarking is its inability to match the detectability levels achieved by **SpARK**, even with optimizations. For example, reducing the key length to 1 in distortion-free methods can slightly improve detectability, but it still falls short of the results achieved by our method. Additionally, **SpARK** offers greater flexibility, allowing users to adjust the trade-off between text quality and detectability by modifying the gamma parameter. This adaptability makes it better suited to diverse use cases, where text quality and watermark robustness may need to be balanced.
> > >
> > > Another significant advantage of **SpARK** is its efficiency in watermark detection. While the text generation times between the two methods are relatively similar, the time required to detect a watermark is significantly lower for **SpARK** compared to distortion-free methods. From the table below, it is evident that the generation time for each of the sample text is comparable between the two methods, but the time it takes distortion-free to detect the watermark per sample is approximately 100 times slower than **SpARK**. This makes our method more practical for large-scale applications where quick detection is essential.
> >
> > || Generation Time (s) | Detection Time (s) |
> > |-|-|-|
> > | Distortion Free | 11.49| 101.59|
> > | **SpARK**| 12.98| 1.11|
> >
> >
> > **Q5:The method proposed in the paper demonstrates a certain level of robustness against lexical substitution attacks. The paper notes that the method effectively maintains robustness when facing a 10% substitution rate, but as the substitution rate increases (e.g., 30% or 50%), the detection accuracy declines. My question is: Do you consider maintaining robustness against a 10% substitution rate as indicative of strong robustness? Additionally, is 10% sufficient for real-world application scenarios?**
> >
> > > **A:** Thank you for your thoughtful question. In our experiments, we evaluated the robustness of watermarking methods under different substitution rates (10%, 30%, and 50%) to determine the threshold at which substitutions impact detectability. While a 10% substitution rate may seem relatively low, it serves as an important benchmark because substitution at higher rates often disrupts the semantic integrity of the sentence as we have discussed in Section 4.3. This is especially true when substitutions are applied randomly, as they can inadvertently alter the meaning or coherence of the text, making results at higher rates less reliable for evaluating robustness in practical contexts.
> > >
> > > For real-world application scenarios, a more realistic substitution method is paraphrasing, such as with tools like DIPPER. Paraphrasing preserves the semantic content while modifying the lexical structure, which closely mirrors how a text might be transformed in actual usage. In these tests, our watermarking method, **SpARK**, demonstrated strong robustness, maintaining high detectability even under paraphrased conditions. In some cases, its performance was close to or at state-of-the-art levels compared to other methods.

---

### Official Review · Reviewer_JRgN · 2024-11-03

**Soundness:** 2
**Presentation:** 2
**Contribution:** 1
**Rating:** 1
**Confidence:** 5

**Summary:**

This paper proposed a sparse text watermarking algorithm via POS (part-of-speech) with the intention to improve the text quality that could be damaged by the KGW algorithm. To be specific, the watermarking algorithm would only be activated when the next token is of a certain group of POS tags. By leaving the generation process of other types of token unchanged, the text quality of generated output can be improved.

**Strengths:**

The idea is simple and easy to understand.

**Weaknesses:**

This paper obviously lacks baselines that are up-to-date to compare with, including but not restricted to SWEET [1], EWD[2]. Simply using the standard KGW[3] and Unigram[4] with different hash functions is far from sufficient in proving the effectiveness of the proposed method. BTW, both SWEET and EWD proposed to selectively perform watermark generation/detection according to token entropy, and have shown better text quality/detection accuracy in their work. To prove the proposed work is better than the listed two, more experiments on text quality, time complexity, detection accuracy and using low-entropy datasets should be conducted.

1. Who wrote this code? watermarking for code generation

2. An Entropy-based Text Watermarking Detection Method

3. A Watermark for Large Language Models

4. Provable Robust Watermarking for AI-Generated Text

**Questions:**

As listed in Weaknesses.

---

> ### Author Response · Authors · 2024-11-22
> **Rebuttal by Authors**
>
> Thank you for the comments, please see our responses below.
>
> **Q1:This paper obviously lacks baselines that are up-to-date to compare with, including but not restricted to SWEET , EWD. Simply using the standard KGW and Unigram with different hash functions is far from sufficient in proving the effectiveness of the proposed method. BTW, both SWEET and EWD proposed to selectively perform watermark generation/detection according to token entropy, and have shown better text quality/detection accuracy in their work. To prove the proposed work is better than the listed two, more experiments on text quality, time complexity, detection accuracy and using low-entropy datasets should be conducted.**
>
> > **A:** Thank you for your suggestions. While SWEET and EWD have demonstrated strong performance in their respective setups, both methods rely on assumptions such as access to the original prompt or the use of a generalized prompt when validated on coding datasets. These assumptions may not always hold in real-world scenarios. In contrast, **SpARK** is designed to work effectively without relying on these assumptions, ensuring broader usability. We also have run some experiments to compare both of these methods to **SpARK** which can be seen in our Public Global Comment.
> In both SWEET[1] and EWD[2], they also tested on low-entropy code generation. However, our paper only focused on high-entropy natural text generation as our method is based on the structure of the natural language to work, and these task are more commonly used compared to code generations.
>
> [1] Lee et al., Who Wrote this Code? Watermarking for Code Generation. ACL 2024
>
> [2] Lu et al., An Entropy-based Text Watermarking Detection Method. 2024.

---

### Official Review · Reviewer_qC2c · 2024-11-04

**Soundness:** 3
**Presentation:** 3
**Contribution:** 2
**Rating:** 5
**Confidence:** 4

**Summary:**

The paper proposes SpARK, a new watermarking method for LLMs that aims to improve detectability and text quality. Unlike traditional methods that watermark every token, SpARK selectively watermarks tokens associated with specific POS tags, such as verbs. By embedding watermarks sparsely and only in specific token positions, SpARK enhances text coherence and robustness against adversarial attacks like synonym substitution and paraphrasing. Experimental results demonstrate that SpARK outperforms other watermarking methods regarding text quality retention and detection accuracy across various tasks and datasets.

**Strengths:**

1. By using POS tags to select tokens for watermarking, SpARK ensures that watermarking is applied selectively and minimally, reducing the model's interference with the natural language structure.

2. The methodology section is well-structured, with algorithms (e.g., POSWatermark, DetectWatermark) clearly defined. SpARK is tested against several existing watermarking techniques (e.g., Hard, LeftHash, SelfHash, and Unigram), allowing for a direct comparison of text quality, semantic similarity, and robustness.

**Weaknesses:**

1. While the method offers an interesting approach, it lacks significant novelty. Similar methods that alter synonyms based on POS tags already exist. This approach instead modifies logits to increase the number of "green" tokens. From a security standpoint, this method has notable weaknesses. In contrast to KGW-type watermarks—which embed watermarks in nearly every token, making it difficult for adversaries to distinguish green from red tokens—this method is more vulnerable. If adversaries know that POS tagging is used in the watermarking process, they could easily defeat it by randomly substituting verbs and nouns with synonyms. Additionally, because POS tagging patterns are relatively consistent across different texts, adversaries would not need to invest much effort to identify and remove the watermark.

2. Although the POS tagging approach is well-motivated, the experiments mainly focus on three tags (Verb, Noun, Determiner), with limited analysis on why these particular tags were chosen over others beyond their document frequency. A broader exploration of different tags and their effectiveness in watermarking would strengthen the study.

**Questions:**

None

---

> ### Author Response · Authors · 2024-11-22
> **Rebuttal by Authors**
>
> Thank you for the comments, please see our responses below.
>
> **Q1: While the method offers an interesting approach, it lacks significant novelty. Similar methods that alter synonyms based on POS tags already exist... Additionally, because POS tagging patterns are relatively consistent across different texts, adversaries would not need to invest much effort to identify and remove the watermark.**
>
> > **A:** In scenarios where the POS tags and seeds of the Sparse Watermark are compromised, it is indeed easier to remove the watermarked tokens compared to dense watermarks. This trade-off, inherent to sparse watermarking, also represents one of its key advantages. We acknowledge this limitation in the paper's Limitation Section and plan to expand on it in the camera-ready version, incorporating further insights from this discussion.
> >
> > To address the vulnerability, one potential mitigation strategy involves dynamically altering the POS tags and seeds used in the watermarking process whenever a leak is detected. The universal POS tagging system provides flexibility for substitution, offering a variety of options to diversify the approach. Furthermore, the method's robustness can be substantially enhanced by employing a combination of multiple POS tags or leveraging the finer-grained Penn Treebank POS tagging scheme [1]. This scheme, with its 36 distinct tags, allows for a more nuanced differentiation of sentence components. By enabling greater precision and complexity in the watermarking process, such adjustments make it significantly more challenging for adversaries to reliably detect or eliminate the watermark.
>
> [1] Taylor et. al, The Penn Treebank: An overview. 2003.
>
> **Q2:Although the POS tagging approach is well-motivated, the experiments mainly focus on three tags (Verb, Noun, Determiner), with limited analysis on why these particular tags were chosen over others beyond their document frequency. A broader exploration of different tags and their effectiveness in watermarking would strengthen the study.**
>
> > **A:** Thank you for your comment. As noted in Appendix Section C.1, we selected three POS tags - Verb, Noun, and Determiner—based on their document frequency of 100%. This choice ensures that these tags consistently appear across all evaluated tasks, providing a stable and robust foundation for watermark application. Additionally, we conducted experiments on these tags and reported their True Positive Rates (TPR) and related metrics in the main paper.
> >
> > To explore the impact of other POS tags on watermark strength and text quality, we extended our analysis to Prepositions/Postpositions, Punctuation, and Adjectives using the same experimental setup on Llama 2. The results demonstrate that these tags achieve high ROUGE-L scores, maintaining the quality of the generated text across both datasets. For Long-Form Question Answering (LFQA), all three tags achieved a TPR higher than 99%. However, for summarization tasks, Adjectives struggled to maintain high TPR due to the shorter answer length, reducing their frequency. Overall, Prepositions/Postpositions and Punctuation are viable for watermarking generated text, but Adjectives may not be as effective for shorter text generation tasks. Below are the results for these tags:
>
> | | LFQA-TPR | LFQA-TNR | LFQA-Rouge L | LFQA-Delta |
> |:-|:-:|:-:|:-:|-|
> | **SpARK** (Preposition/Postposition) | 100%| 98%| 20.45| -4.85%|
> | **SpARK** (Punctuation)| 100%| 99.25%| 18.46| -13.33%|
> | **SpARK** (Adjective)| 100%| 99.5%| 20.34| -5.325%|
>
> | | Sum-TPR | Sum-TNR | Sum-Rouge L | Sum-Delta |
> |:-|:-:|:-:|:-:|-|
> | **SpARK** (Preposition and Postposition) | 100%| 98.25%  | 21.53| -8.26%|
> | **SpARK** (Punctuation)| 99.25%  | 98.25%  | 19.59| -16.53%   |
> | **SpARK** (Adjective)| 95.5%   | 99.25%  | 22.43| -4.43%|
>
> > Furthermore, we conducted paraphrasing attack experiments using the DIPPER to evaluate the robustness of these additional POS tags under adversarial conditions. Our findings revealed that while all tested tags exhibited some level of detectability after paraphrasing, their robustness varied significantly. Among the three, only the Punctuation tag demonstrated detectability on par with our baseline methods, showcasing its potential as a reliable choice for watermarking even in the presence of paraphrasing attacks. We recognize the value of this analysis in advancing the robustness of our method and will include these findings, along with a detailed discussion, in the camera-ready version of the paper to provide a more comprehensive evaluation.
>
> | |   40O  | 40O-40L |
> |-|:-|:-:|
> | **SpARK** (Adjective)|  31.75 |  22.25   |
> | **SpARK** (Preposition and Postposition) |  47.50 |  34.00   |
> | **SpARK** (Punctuation)|  64.38 |  47.00   |

---

### Author Response · Authors · 2024-11-22
**Global Comment for additional experiment.**

>To ensure a fair comparison, we evaluated EWD and SWEET on the WaterBench baseline under the same experimental conditions as described in the main paper, but without providing access to the original prompt.  From the results, we can see that while both EWD and SWEET achieve higher text quality compared to some other watermarking methods, **SpARK** consistently outperforms EWD in terms of Rouge-L scores across tasks. This indicates that **SpARK** offers a better balance between quality and detection performance, while not needing additional information.


|                     | LFQA-TPR | LFQA-TNR | LFQA-Rouge L | LFQA-Delta |
|:-------------------|:--------|:--------|:------------|:----------|
| EWD                 | 100%     | 100%     | 18.12        | -16.07%    |
| SWEET               | 99.8%    | 100%     | 18.88        | -12.55%    |
| **SpARK** (Verb)       | 100%     | 99%      | 18.87        | -12.60%    |
| **SpARK** (Noun)       | 100%     | 99.5%    | 18.48        | -14.40%    |
| **SpARK** (Determiner) | 100%     | 98.8%    | 19.20        | -11.07%    |


|                     | Sum-TPR | Sum-TNR | Sum-Rouge L | Sum-Delta |
|---------------------|---------|---------|-------------|-----------|
| EWD                 | 100%    | 100%    | 18.11       | -22.83%   |
| SWEET               | 100%    | 100%    | 18.13       | -22.71%   |
| **SpARK** (Verb)       | 100%    | 99.5%   | 20.95       | -10.74%   |
| **SpARK** (Noun)       | 100%    | 99.5%   | 18.39       | -21.64%   |
| **SpARK** (Determiner) | 100%    | 98.0%   | 20.89       | -10.99%   |


> In addition, we tested the robustness of EWD and SWEET under paraphrasing attacks using the DIPPER as the paraphrasing model. As we can see from the table below, the detectability of EWD is less than **SpARK** after being paraphrased. This shows that **SpARK** maintained higher robustness to paraphrasing attacks, highlighting its effectiveness in adversarial scenarios. On the other hand, while SWEET demonstrates high robustness against paraphrasing attacks, **SpARK** can still maintain higher detectability among the sparse watermark, while generating higher-quality text, striking a balance between adversarial robustness and text quality.

|                     |  40O | 40O-40L |
|:-------------------|:----:|:-------:|
| EWD                 | 52.7 | 38.1    |
| EWD (with prompt)   | 52.1 | 40.1    |
| SWEET               | 69.1 | 54.5    |
| SWEET (with prompt) | 69.1 | 54.5    |
| **SpARK** (Verb)       | 54.3 | 43.5    |
| **SpARK** (Noun)       | 53.9 | 41.9    |
| **SpARK** (Determiner) | 74.3 | 66.9    |

>Lastly, we assessed the computational efficiency of each method. While **SpARK** has a higher detection time compared to other watermark methods, it is still a viable option compared to other published methods that have detection time in minutes. Not to mention that both EWD and SWEET also needed access to the original LLM in order for it to detect the watermarks.

|           | Generation Time (s) | Detection Time (s) |
| --------- | ------------------- | ------------------ |
| SelfHash  | 25.92               | 0.03               |
| LeftHash  | 6.46                | 0.11               |
| Hard      | 6.45                | 0.13               |
| Unigram   | 6.45                | 0.003              |
| EWD       | 7.10                | 0.2                |
| SWEET     | 6.39                | 0.18               |
| **SpARK** | 7.89                | 1.01               |


>In conclusion, although it can incur slight overhead during detection, **SpARK** offers a more robust, reliable, and versatile solution for watermarking tasks. It outperforms EWD and SWEET in detection accuracy and text quality while staying resilient under constrained and adversarial scenarios, ensuring practical applicability in diverse real-world environments.

---

### Author Response · Authors · 2024-11-25
**Thank you for your valuable comments and welcome additional questions!**

We want to start by thanking the reviewers once more for providing constructive feedback and raising questions to help us improve the quality of our paper. We have addressed all the questions and suggestions from the reviewers in our rebuttal with new experiments as recommended. Please share your additional thoughts and we're happy to address them promptly.

---

### Author Response · Authors · 2024-12-01
**We welcome additional questions!**

We hope this message finds you well. As a gentle reminder, the deadline for providing your additional feedback is today. We greatly appreciate the time and effort you’ve dedicated to reviewing our work and providing your constructive feedback. However, since our previous reminder, we have not yet received any additional responses from reviewer. If any reviewers have any further questions or suggestions, please don’t hesitate to share them with us—we are committed to addressing them promptly.

---

### Author Response · Authors · 2024-12-04
**Summary of the rebuttal.**

Thank you everyone for your helpful comments. In this work, we introduce **SpARK**, a novel watermarking method for large language models. With extensive evaluations, our method achieves state-of-the-art generation quality while also offering competitive robustness against attacks in comparison with other watermarks.

During the rebuttal process, we have:

- Provided additional experiments on other quality-preserving watermark methods, such as SWEET and EWD.
- Provided additional experiment results on different POS tags,  a wider exploration of different tags, and their effectiveness in watermarking.
- Provided additional analysis regarding the advantages of using SpARK compared to Distortion-free watermark.
- Provided additional comparisons with other watermark methods on generating overhead and detection time.
- Provided references to sections in our paper to answer the reviewers' questions.

We hope that our responses have appropriately addressed the questions of the reviewers. **SpARK** is highly applicable with the advantages in producing both a resilient watermark and cohesive text. If the reviewers have any additional comments or questions, we are more than happy to answer.

---

### Meta-Review · Area_Chair_HgP8 · 2024-12-16

**Metareview:**

The paper introduces SpARK, a sparse watermarking method for large language models (LLMs) that selectively embeds watermarks in tokens associated with specific part-of-speech (POS) tags, such as verbs. By targeting only certain POS tags, SpARK minimizes disruptions to the text generation process, preserving text quality while maintaining high watermark detectability. This selective approach also enhances robustness against adversarial attacks like synonym substitution and paraphrasing. Experimental results demonstrate that SpARK outperforms traditional watermarking methods in detection accuracy and text coherence. However, the paper raises questions about the novelty of this approach, as sparse watermarking is a well-explored concept.

The proposed method is simple yet effective. However, as pointed out by the reviewers, the topic of LLM watermarking is currently very popular, and there are numerous similar methods already available [1]. As a result, the novelty of the proposed method appears to be limited. I encourage the authors to review the existing literature, particularly surveys on LLM watermarking, and revise the paper based on the reviewers’ feedback before considering resubmission to a future venue.

[1] Watermarking Techniques for Large Language Models: A Survey
https://arxiv.org/pdf/2409.00089

**Additional Comments On Reviewer Discussion:**

Most of the reviewers raised concerns about the novelty of the proposed method. While the authors have addressed some of these concerns, there are still issues that remain unresolved. I also noted that one reviewer gave a strong rejection, which I believe may have been somewhat excessive. Although I assigned less weight to that particular review, the average scores from the other reviewers are still low, making the paper unsuitable for acceptance at ICLR. Therefore, I recommend rejecting the paper.

---

### Decision · Program_Chairs · 2025-01-22

Reject